# Local CO$_2$ reservoir layer promotes rapid and selective electrochemical CO$_2$ reduction

Subhabrata Mukhopadhyay[1], Muhammad Saad Naeem [2,3], G. Shiva Shanker[1], Arnab Ghatak [1], Alagar R. Kottaichamy[1], Ran Shimoni[1], Liat Avram [4], Itamar Liberman[1], Rotem Balilty[1], Raya Ifraemov[1], Illya Rozenberg[1], Menny Shalom [1], Núria López [2] ✉ & Idan Hod[1] ✉

Electrochemical CO$_2$ reduction reaction in aqueous electrolytes is a promising route to produce added-value chemicals and decrease carbon emissions. However, even in Gas-Diffusion Electrode devices, low aqueous CO$_2$ solubility limits catalysis rate and selectivity. Here, we demonstrate that when assembled over a heterogeneous electrocatalyst, a film of nitrile-modified Metal-Organic Framework (MOF) acts as a remarkable CO$_2$-solvation layer that increases its local concentration by ~27-fold compared to bulk electrolyte, reaching 0.82 M. When mounted on a Bi catalyst in a Gas Diffusion Electrode, the MOF drastically improves CO$_2$-to-HCOOH conversion, reaching above 90% selectivity and partial HCOOH currents of 166 mA/cm$^2$ (at −0.9 V vs RHE). The MOF also facilitates catalysis through stabilization of reaction intermediates, as identified by operando infrared spectroscopy and Density Functional Theory. Hence, the presented strategy provides new molecular means to enhance heterogeneous electrochemical CO$_2$ reduction reaction, leading it closer to the requirements for practical implementation.

It is environmentally essential to reduce the dependence on fossil fuels and develop a sustainable alternative[1–3]. A crucial step to achieve this goal is to circumvent renewable energy resources' intermittency by creating reliable energy storage mechanisms. *Carbon capture, utilization, and storage* in the form of solar fuel is among one of the most acknowledged means to reduce the carbon footprint and limit the global temperature increase[3,4]. Electrocatalytic CO$_2$ reduction reaction (eCO$_2$RR) to generate valuable chemicals and high energy density fuels provides a potential route to realize the utilization of CO$_2$ and store renewable energy in usable form[5–8]. However, the critical challenge is developing a stable, efficient, and economic catalyst which can perform eCO$_2$RR with high activity and selectivity[1]. Over the last two decades significant progress was achieved in developing earth abundant metal catalysts for eCO$_2$RR [9–12]. Multiple proton-coupled electron transfers can lead to a large variety of reduced carbon products as

CO[13,14], HCOOH[15–18], CH$_4$[19–21], as well as multi-carbon products[9,10,22,23]. Despite the great progress achieved so far, one of the remaining bottlenecks is the poor solubility of CO$_2$ in aqueous solutions (~30 mM)[24], which limits the availability of CO$_2$ near the heterogeneous catalytic sites and thus lowers the overall efficiency of the process by promoting the competitive hydrogen evolution reaction (HER).

To control the reaction near the electrode surface, several reports have explored the effect of catalyst's surface modification with polymers, organic ligands, and hybrid organic-inorganic layers to suppress HER and facilitate more efficient eCO$_2$RR[12,22,25,26]. We realized that due to their ultra-high porosity and structural modularity[27,28], Metal-Organic Frameworks (MOFs) could serve as a functional mimic of metalloenzymes and membrane-proteins, that enable precise modulation of catalytic operation course of a catalysis without affecting the nature of the actual catalytically active site[29–31]. In principle, a suitably

[1]Department of Chemistry and Ilse Katz Institute for Nanoscale Science and Technology, Ben-Gurion University of the Negev, Beer-Sheva 8410501, Israel. [2]Institute of Chemical Research of Catalonia (ICIQ-CERCA), The Barcelona Institute of Science and Technology (BIST), 43007 Tarragona, Spain. [3]Universitat Rovira i Virgili, Pl. Imperial Tarraco 1, 43005 Tarragona, Spain. [4]Department of Chemical Research Support Weizmann Institute of Science, Rehovot 7610001, Israel. ✉e-mail: nlopez@iciq.es; hodi@bgu.ac.il

designed MOF-membrane can improve electrocatalysis rate and selectivity by (i) pre-concentrating a desired substrate (e.g., $CO_2$) at close proximity of the catalytic center, and (ii) providing secondary sphere functionalities to accelerate catalysis[8,32]. Yet, although MOFs have been extensively studied for various potential applications[33–36], including electrocatalysis[37–41], they were rarely employed to impose molecular-level control over heterogeneous catalytic operation[8].

In this work, we prepared a MOF-membrane which can selectively pre-concentrate and activate $CO_2$ prior to its electroreduction by the underlying solid electrocatalyst. Specifically, encouraged by the fact that $CO_2$ solubility in acetonitrile (0.28 M) is one order of magnitude higher than in pH 7 aqueous solutions (-0.03 M), we have post-synthetically modified the MOF pores of a $Zr_6$-oxo based UiO-66 MOF[42] with nitrile functional groups (termed UiO-66-CN). To study the effect of modified UiO-66-CN MOF-membrane on $eCO_2RR$ performance, we took the well-known $CO_2$-to-HCOOH conversion on Bi electrocatalyst[15,17,18,43,44], as a model system (Fig. 1).

In a conventional H-cell configuration, a thick UiO-66-CN membrane mounted on a Bi-foil enhanced $eCO_2RR$ rate by 7-fold, while improving HCOOH selectivity to 93% (at −0.75 V vs. RHE). More importantly, in a Gas-Diffusion Electrode (GDE) setup, application of the practically insulating MOF layer results in an impressive enhancement of catalytic current, reaching 166 mA/cm[2], and HCOOH selectivity (up to 98%). Infrared Reflection Absorption Spectroscopy (ATR-IRRAS) combined with Density Functional Theory (DFT) simulations reveal that the UiO-66-CN membrane enhances $eCO_2RR$ activity by increasing local $CO_2$ solubility (0.82 M) and stabilizing catalytic intermediates.

## Results and discussion

For this work, the $eCO_2RR$ catalyst was prepared by depositing a porous membrane of a modified UiO-66 MOF over catalytically active Bi foil. The MOF-membrane was grown from a UiO-66 gel precursor. The synthesis of the gel was adapted with modifications from earlier works of Bueken et al. [42].

The UiO-66-CN gel was synthesized from the as-synthesized UiO-66 gel by solvent-assisted ligand incorporation (SALI) with cyano-benzoic acid (BA-CN) ligand[45]. Thereafter, the MOF gel precursors were drop-casted on Bi-electrodes to form the Bi-UiO-66 membranes

(See method section and Table S1 for detailed synthetic methodology of UiO-66 gel, UiO-66-CN gel and the porous membranes on Bi foil).

Powder X-ray diffraction (PXRD) analysis confirmed that the UiO-66 membranes were successfully synthesized over Bi foil (Fig. 2a). Scanning electron microscopy (SEM) analysis (Fig. S3) and SEM-FIB (FIB=Focused Ion Beam) analysis (Fig. 2b–d) show that the Bi-UiO-66 membranes are porous, homogeneous, and continuous (See method section, for experimental techniques and calculations). Different amounts of UiO-66 gel precursor were used to synthesize three distinct UiO-66 membranes on top of Bi-foil, namely, Bi-UiO-66-A, Bi-UiO-66-B and Bi-UiO-66-C. As determined by the SEM-FIB analysis, the membrane thicknesses for Bi-UiO-66-A, Bi-UiO-66-B and Bi-UiO-66-C were 14.5 μm, 18.4 μm, and 32.5 μm, respectively.

X-ray photoelectron spectroscopy (XPS) was performed to understand the chemical nature of the Bi-UiO-66 systems. The survey scan confirmed the presence of the Zr, C, and O (Fig. S4a), while a peak for Bi appeared for Bi-UiO-66-A-C after sample etching with ion-beam (Fig. S4b). Thus, SEM-FIB and XPS analysis indicate homogeneous coating of UiO-66 membrane for Bi-UiO-66-A-C systems. A combination of inductively coupled plasma optical emission spectrometry (ICP-OES) and [1]H-NMR was utilized to determine the chemical composition of the UiO-66 membranes (Figs. S5–S8). Here it can be expected that, since we had used the same precursor UiO-66 gel to grow the three UiO-66 membranes i.e., Bi-UiO-66-A-C, the chemical composition of all membranes is identical. The number of benzene-1,4-dicarboxylate (BDC) linkers per $Zr_6$-oxo node is 8, instead of 12 for an ideal UiO-66 structure (see SI section S1).

This structural information was employed to build the interface models for DFT simulations (Fig. 3, see method section and SI section S1, Tables S2–S6 for details). The $Zr_6$-oxo nodes (capped by H atoms to reduce the computational cost) were taken as the building unit for UiO-66-CN (Fig. 3b). The missing linker sites of UiO-66 were key to incorporate the nitrile groups and prepare UiO-66-CN. In the DFT models, the BA−CN linker of Bi-UiO-66-CN replaced one of the terminal -OH of the UiO-66. Next, UiO-66-CN was attached to Bi(0001) surface by cutting one of the MOF core planes of UiO-66 with carboxylate centers, allowing the underlying oxygens to interact with the Bi surface, resulting in Bi-UiO-66-CN. The same procedure was used to construct Bi-UiO-66 by changing the carboxylate of the BDC linker

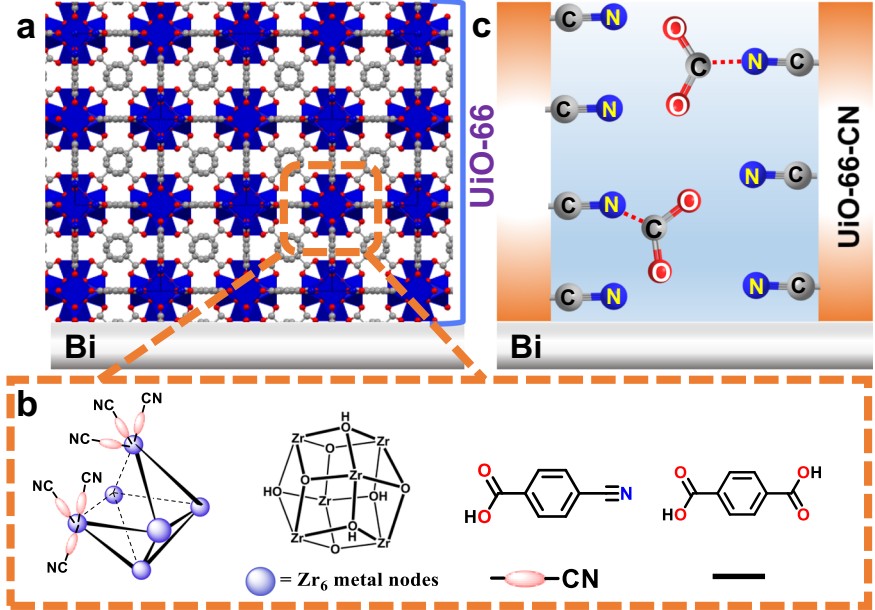

**Fig. 1 | Illustration of the structure of a UiO-66-CN membrane. a** Assembled on a Bi electrocatalyst. **b** Illustration of a cyano-benzonic acid (BA-CN) attachment at missing-linker defect sites in UiO-66 membrane. **c** Illustration of increased local $CO_2$ concentration at the vicinity of the Bi electrocatalyst surface.

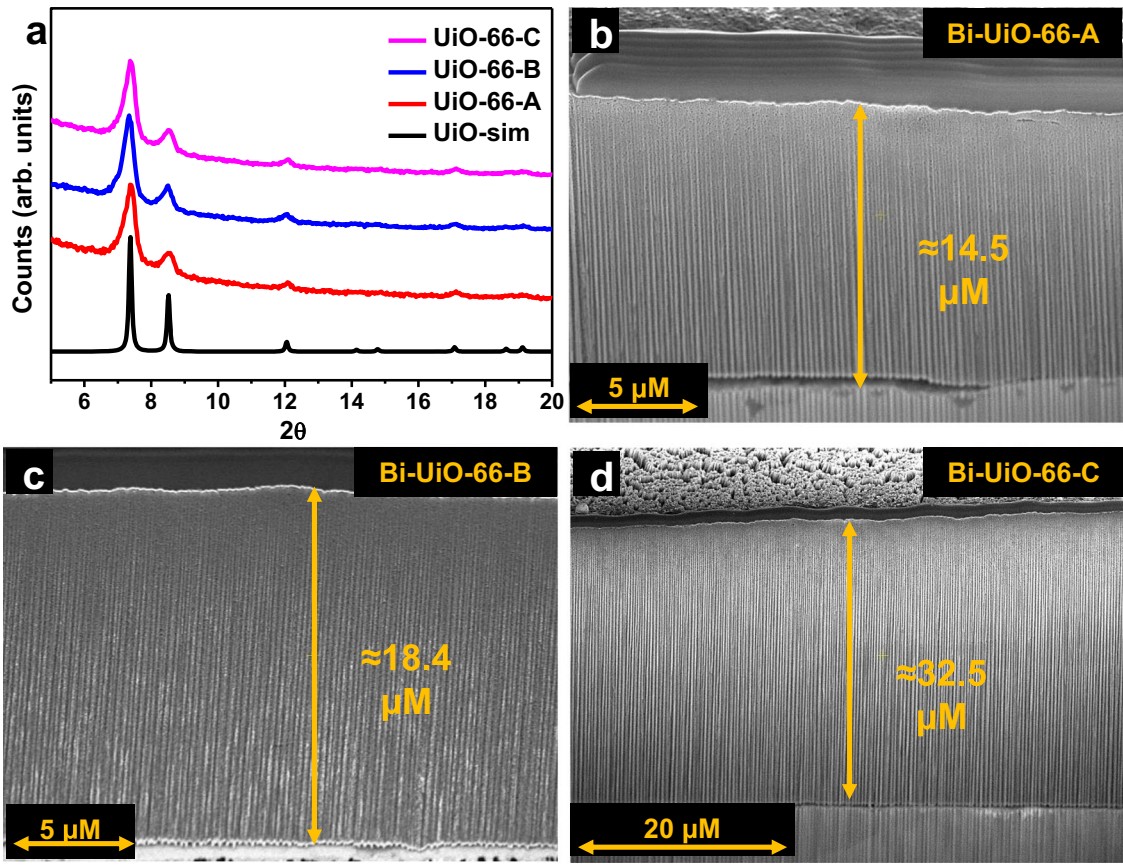

**Fig. 2 | Characterizations of Bi-UiO-66-A-C. a** PXRD patterns of Bi-UiO-66-A-C with the simulated pattern of UiO-66 (UiO-sim). SEM-FIB cross-section images of **b** Bi-UiO-66-A, **c** Bi-UiO-66-B, and **d** Bi-UiO-66-C.

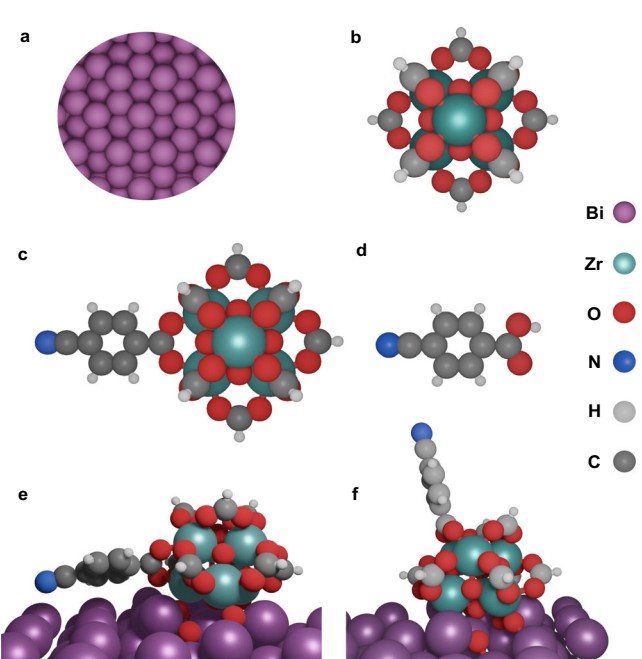

**Fig. 3 | The DFT model structures illustrated. a** Bismuth (0001) surface, **b** Zr$_6$-oxo UiO-66 node, **c** nitrile modified Zr$_6$-oxo node (UiO-66-CN), **d** BA-CN linker, **e** Bi-UiO-66-CN, when BA-CN is parallel to Bi (0001) plane (parallel mode), **f** Bi-UiO-66-CN$_\perp$, when BA-CN is perpendicular to Bi (0001) plane. Complementary alternative models are presented in SI (Table S7).

with an H atom instead of CN group (see method section and SI section S1). To account for the different probable orientations of the BA-CN, two modes were accounted for: (a) parallel to Bi-foil (Fig. 3e) and perpendicular to Bi-foil, Bi-UiO-66-CN$_\perp$ (Fig. 3f). Complementary alternative models were built, and their major fingerprints investigated, see SI (Table S7). The model presented here is the most compatible with all the experimental observations.

eCO$_2$RR measurements were carried out in a gas-tight H-cell, in CO$_2$-saturated 0.1 M NaHCO$_3$ solution (pH 6.8), using Ag/AgCl (saturated KCl), Platinum foil, and Bi-foil (with and without UiO-66-A-C membrane) as the reference, counter, and working electrodes, respectively (see SI method section for details). The eCO$_2$RR performance of bare Bi-foil and Bi-UiO-66-A-C was determined by performing bulk electrolysis at every 50 mV intervals within −0.6 V to −0.9 V vs. RHE. Although there are a handful of recent reports of efficient eCO$_2$RR by Bi-based catalysts, flat Bi-foil is generally not considered efficient for CO$_2$ to HCOOH conversion and HER is dominant during electrocatalysis. Indeed, HCOOH and H$_2$ were determined as the major eCO$_2$RR products, along with a trace amount of CO ( < 2%). The combined Faradaic efficiencies of HCOOH (FE$_{HCOOH}$), hydrogen (FE$_{H2}$), and CO (FE$_{CO}$) were essentially 100% (Fig. S9).

FE$_{HCOOH}$ for the Bi-UiO-66-A-C (Bi-foil with UiO-66-A-C membrane) were significantly higher than that of Bi-foil (Fig. 4a). With increasing membrane thickness, the HCOOH production could be systematically varied. At −0.75 V vs RHE, FE$_{HCOOH}$ for Bi-foil, Bi-UiO-66-A, Bi-UiO-66-B and UiO-66-C were 20%, 54%, 80%, and 66%, respectively. Thus, Bi-UiO-66-B exhibited the best HCOOH selectivity (Fig. 4a). All UiO-66-A-C membranes are essentially structurally similar as they were grown from the same parent UiO-66 gel. Consequently, the differential HCOOH selectivity of UiO-66-A-C depends only on the

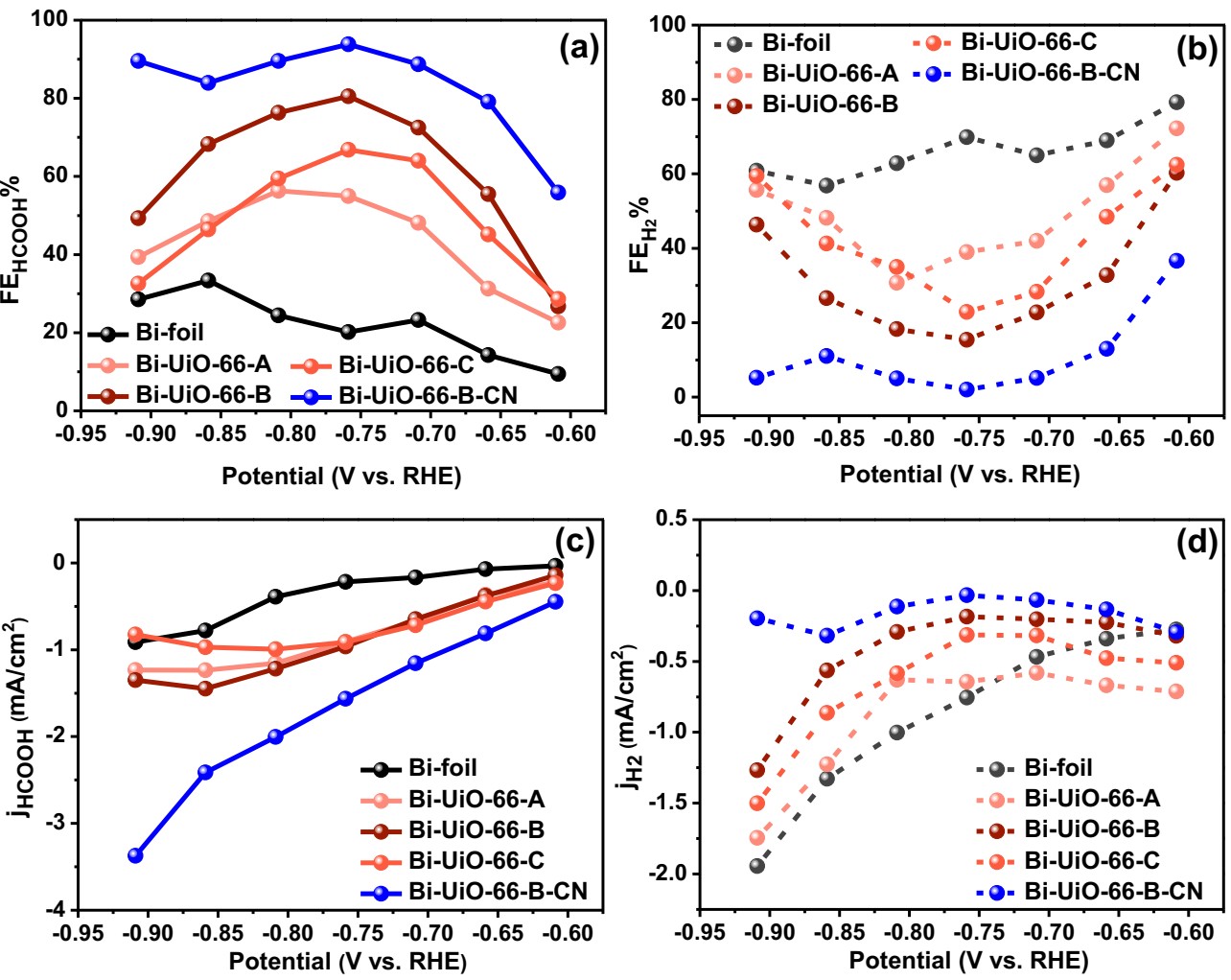

**Fig. 4 | Variation in Faradaic efficiencies (FE) of Bi-foil and UiO-66-A-C membrane containing Bi-foil. a** FE for HCOOH and **b** FE for $H_2$ at different applied potentials. Variation of partial catalytic currents of **c** HCOOH production ($j_{HCOOH}$) and **d** $H_2$ evolution reaction ($j_{H2}$) at different applied potentials.

variation in attenuation of diffusional mass transport through the UiO-66 membranes[31]. Therefore, the thickness of UiO-66-B was chosen as optimum for the rest of our study.

In the next step, a UiO-66 gel was post-synthetically modified with BA-CN and was subsequently used to grow a nitrile-functionalized UiO-66-B membrane (UiO-66-B-CN). Diffused reflectance infrared Fourier transform spectroscopy (DRIFTS) was utilized to confirm a successful immobilization of BA-CN on the MOF's $Zr_6$-oxo nodes (Fig. S10a). In UiO-66-CN, the BA-CN replaces the terminal −OH groups of $Zr_6$-oxo nodes, causing an overall shift in its IR peak from 3674.8 $cm^{-1}$ for UiO-66 to 3671.0 $cm^{-1}$ for UiO-66-CN[45]. The UiO-66-B-CN thickness was determined by SEM-FIB analysis to be 18.3 μm (Fig. S10b). Moreover, the nature of the Bi catalyst was unaffected by the growth of UiO-66-B-CN membrane, as confirmed via Bi 4f XPS analysis (Fig. S11, Table S9).

As seen in H-cell experiments, $FE_{HCOOH}$ was significantly enhanced to ~93% for Bi-UiO-66-B-CN (Fig. 4a). Importantly, Bi-UiO-66-B-CN exhibits $j_{HCOOH}$ enhancement of more than 7 times compared to bare Bi-foil (at −0.75 $V_{RHE}$), while its partial current density due to HER ($j_{H2}$) was drastically lowered compared to the other samples (Fig. 4c, d). Meaning, the nitrile-modified MOF (Bi-UiO-66-B-CN) significantly increase the kinetics of $eCO_2RR$ and supresses HER, and thus also leads to its substantial improvement of the overall selectivity toward HCOOH. Supressed HER in presence of the MOF-based membranes was further confirmed by performing linear sweep voltammetry (LSV) under Ar environment (Fig. S12).

In addition, Electrochemical Impedance Spectroscopy (EIS) analysis was conducted for the different samples under catalytic operation potentials. As seen in Fig. S13a, Bi-UiO-66 and Bi-UiO-66-CN exhibit reduced resistance for charge transfer at the Bi-electrolyte interface compared to bare Bi-foil, thus signalling for acceleration of catalysis rate (e.g. higher catalytic currents) for the MOF-coated samples. In addition, compared to bare Bi-foil, UiO-66 coated Bi shows higher resistance for diffusional mass transport in solution (in accordance with attenuation of diffusional mass transport through the UiO-66 membranes). Yet, Fig. S13b reveals that UiO-66-CN coated Bi exhibits similar mass transport resistance as Bi foil, presumably due to the high local $CO_2$ concentration within the UiO-66-CN membrane, thus enabling larger reactant delivery toward the catalytic surface (see detailed discussion in SI).

Next, we were set to test the MOF-membrane approach in a more practical electrochemical setup. First, we have examined the impact of UiO-66-CN membrane on $eCO_2RR$ performance of a high surface area porous Bi electrode (p-Bi) using conventional H-cell set-up (method section and Figs. S14–S22). The enhancement of $eCO_2RR$ by UiO-66-CN in terms of FE was similar for p-Bi and flat Bi electrodes, but with significantly higher catalytic current (up to 30 mA/$cm^2$).

The UiO-66-CN modified electrode was then tested in a GDE flow cell. To do so, we have deposited Bi NPs catalyst on 20% PTFE coated Gas diffusion layer (GDL) (Fig. 5a) and characterized it using XRD and SEM (Fig. S23). UiO-66-CN membrane was grown atop the Bi-GDE as a

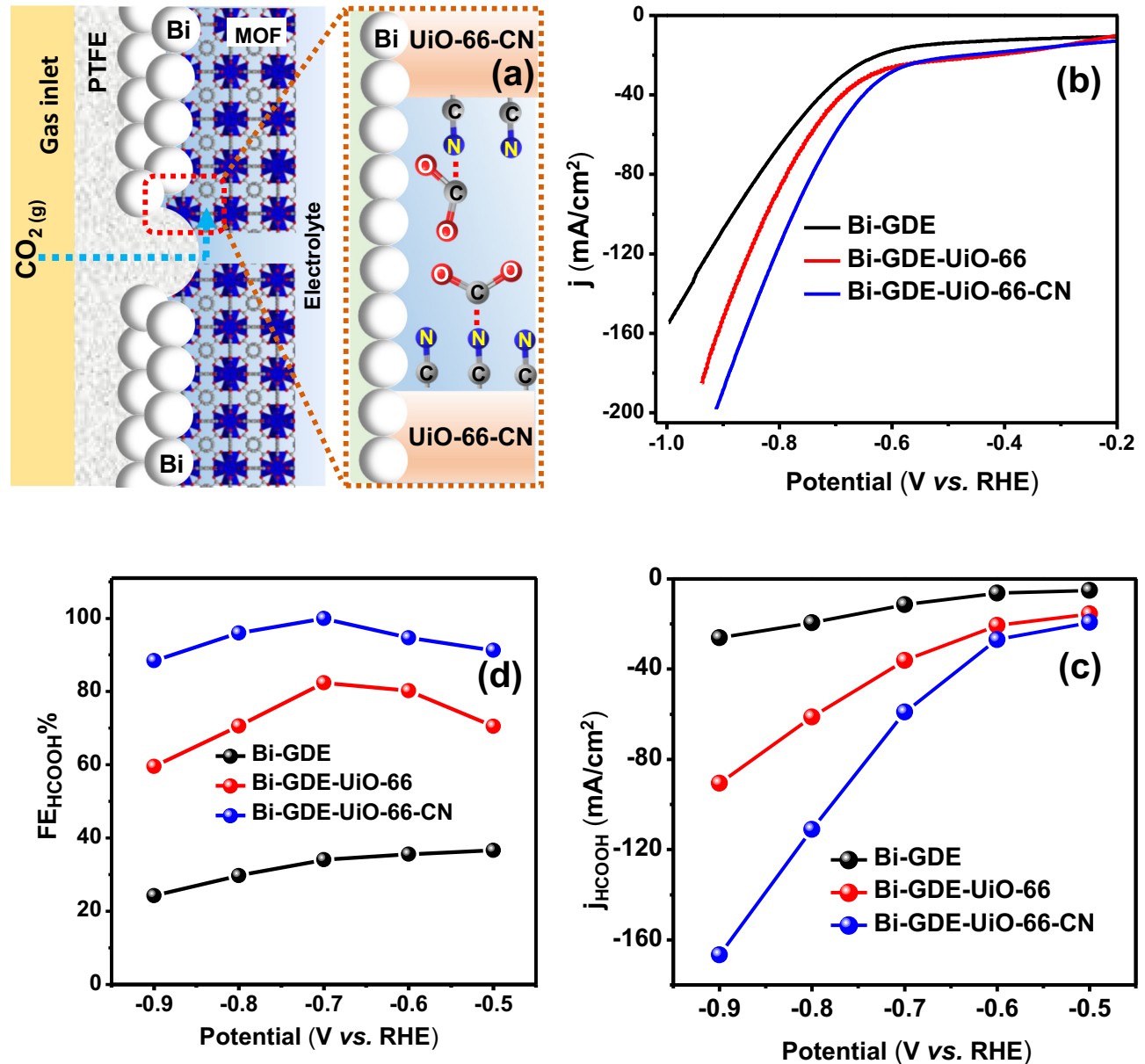

**Fig. 5 | Schematic representation of Bi-GDE-UiO-66-CN gas-diffusion electrode (GDE) and its electrochemical CO₂ reduction performance. a** Schematic structural representation of Bi-GDE-UiO-66-CN GDE highlighting the functional properties of UiO-66-CN membrane as $CO_2$ solvation layer in the GDE set-up. **b** Linear sweep voltammograms (LSVs), **c** Variation in Faradaic efficiency ($FE_{HCOOH}$%) of HCOOH production, and **d** variation of partial current density for HCOOH formation ($j_{HCOOH}$) with potential for Bi-GDE, Bi-GDE-UiO-66 and Bi-GDE-UiO-66-CN.

continuous MOF membrane (see method section) and thoroughly characterized (Fig. S24). The electrochemical CO₂ reduction performance (Fig. 5 and Fig. S25) of the Bi-GDE was compared with the MOF-modified Bi membrane (Bi-GDE-UiO-66 and Bi-GDE-UiO-66-CN). Figure 5 demonstrates that Bi-GDE-UiO-66-CN improves both total currents and HCOOH selectivity, reaching HCOOH partial currents of 166 mA/cm² (at −0.9 V vs RHE). This current is more than 6-times higher compared to only Bi-GDE (Fig. 5d) and is comparable to the state-of-the-art electrochemical CO₂-to-HCOOH conversion systems in bicarbonate-based electrolytes (see comparison Table S8).

We have also checked the stability of the Bi-GDE-UiO-66-CN system by 25 h of continuous chronoamperometric measurement (Fig. S26) and after the prolonged electrolysis by XRD and SEM, (Figs. S27, S28). The catalyst system was highly durable and maintained a high catalytic performance of more than 80% till the end of the measurement. Material characterization essentially shows that the

MOF membrane structure and morphology is retained. The obtained results emphasize the fact that the capacity of the MOF membrane to assist the eCO₂RR performance is equally applicable for a GDE with high catalytic current density as it can be effective for a flat Bi-catalyst.

By now we realize that the increased electrocatalytic activity and selectivity of UiO-66-B-CN was achieved by exploiting the capacity of the CN-modified membrane to tune the microenvironment surrounding the Bi surface. Thus, we were interested to uncover the mechanisms that govern the system's catalytic operation. First, to monitor the extent of CO₂ solvation by UiO-66-B-CN, ATR-IRRAS (in Otto configuration) analysis was performed in a CO₂-purged 0.1 M NaHCO₃ solution for UiO-66-B and UiO-66-B-CN membranes coated surfaces (see method section). During a CO₂ purge cycle, the ATR-IRRAS spectrum of UiO-66-B exhibits peaks for gas-phase CO₂ located at 2360 cm⁻¹ and 2332 cm⁻¹ (Fig. 6a), commonly present in CO₂ purged aqueous solution. Another peak arises at 2345 cm⁻¹, corresponding to

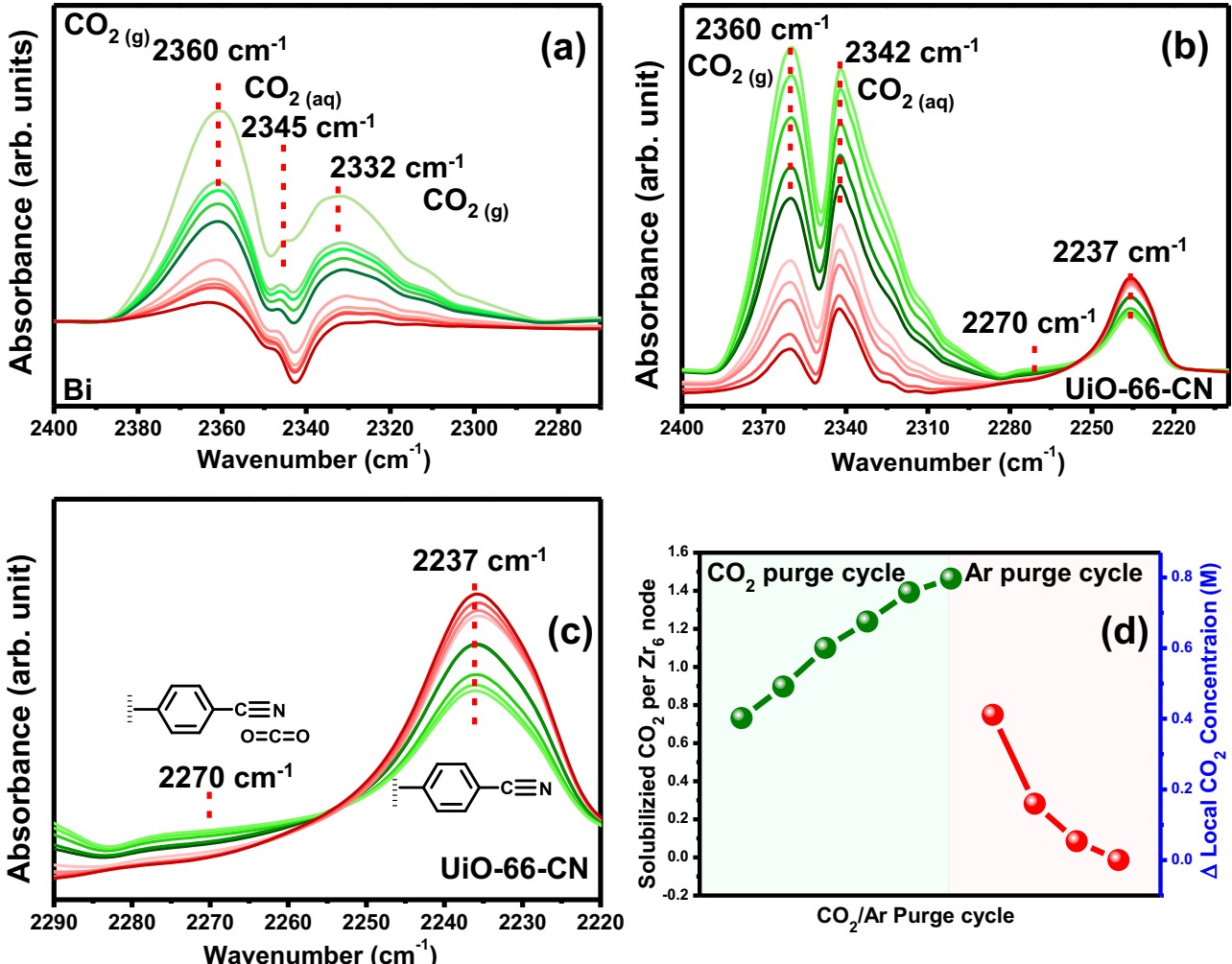

**Fig. 6 | ATR-IRRAS in Otto configuration spectra focusing on the extent of CO₂ solubilization.** CO₂ solubilization in the presence of **a** UiO-66 membrane and **b** UiO-66-CN membrane. **c** Zoomed in ATR-IRRAS spectrum of UiO-66-CN depicting the changes in the peak intensity of BA-CN along with CO₂ concentration change.

The effect of increasing and decreasing CO₂ concentration was studied by purging CO₂ and Ar in cycles in the aqueous solution. **d** Variation of the CO₂ solubilization per Zr₆-oxo nodes and increase in local CO₂ concentration because of UiO-66-B-CN membrane during CO₂ and Ar purge cycles.

the highly solvated (dissolved) $CO_2$ as was reported earlier for solvated $CO_2$ in saline aqueous solution[24]. Varying the $CO_2$ concentration in the solution (either by the extent of $CO_2$ purge or by purging with Ar), changes the solvated $CO_2$ peak's intensity, indicating the reversibility of the $CO_2$ interaction with the membrane. Unlike UiO-66-B, for UiO-66-B-CN, a gas-phase $CO_2$ peak appears at 2360 cm⁻¹, alongside a major peak for solvated $CO_2$ at ~2342 cm⁻¹ (Fig. 6b). Remarkably, the high intensity of the 2342 cm⁻¹ peak, masks the second gas-phase $CO_2$ peak at 2332 cm⁻¹ for UiO-66-B-CN, indicating the high efficacy of UiO-66-B-CN to solubilize $CO_2$. As seen in Table S2, the DFT change in frequency is 2366 cm⁻¹ to 2353 cm⁻¹ for gas-phase $CO_2$ to $CO_2$ adsorbed by Bi-UiO-66-CN. Such shift of 13 cm⁻¹ between gas-phase $CO_2$ and $CO_2$ adsorbed to Bi-UiO-66-CN is in reasonable agreement with the experimental measurements (18 cm⁻¹) and can be assigned to the combined interaction with CN and the Bi (Table S2). The frequency shift, combined with the bending of $CO_2$ (1.8°) implies the activation of $CO_2$ by the nitrile groups. The $CO_2$ activation using Bi-UiO-66-CN can further be visualized through charge density difference plots (Fig. S29). Further, the relative ATR-IRRAS peak intensities for dissolved $^{13}CO_2$ (2270 cm⁻¹) and gaseous $^{13}CO_2$ (2290 cm⁻¹) were compared for UiO-66-B and UiO-66-B-CN (Fig. S30). The peak intensity at 2270 cm⁻¹ attributed to dissolved $^{13}CO_2$ was higher for UiO-66-B-CN than UiO-66-B. The observation supports the earlier conclusion of enhancement of $CO_2$

solvation by UiO-66-CN in aqueous solution (see detailed discussion in SI). ATR-IRRAS measurements were also performed for an aqueous solution of BA-CN under similar conditions (Fig. S31 and SI, section S2). The relative intensity for gaseous and dissolved $CO_2$ peak for BA-CN were similar to that of UiO-66-B-CN (Fig. S31). Thus, it is evident that the BA-CN introduced in the nitrile-modified MOF-membrane plays a pivotal role in increasing local $CO_2$ solvation.

Interestingly, for both solution-dissolved BA-CN and UiO-66-B-CN, the IR peak corresponding to the nitrile C – N stretch at 2237 cm⁻¹ decreases gradually with increased $CO_2$ concentration, while a new broad band arises at ~2270 cm⁻¹ (Fig. 6c and S30). Meaning, the nitrile groups of UiO-66-B-CN can solvate and activate $CO_2$, and thus lower the relative abundance of non-interacting nitriles. Hence, the peak intensity for the non-interacted nitriles (at 2237 cm⁻¹) decreases with the solubilization of $CO_2$, while the newly formed band at ~2270 cm⁻¹ corresponds to the $CO_2$-coordinated nitriles as also suggested by DFT (Fig. 6c and Fig. S31). As such, since we know the total amount of nitrile groups per Zr₆-oxo MOF-node, the integration of Bi-UiO-66-B-CN's 2237 cm⁻¹ peak area under different $CO_2$ concentrations provides an indirect way of quantifying the extent of local $CO_2$ solubilization (Fig. 6d, SI section S2). The number of $CO_2$ molecules solubilized per Zr₆-oxo nodes of UiO-66-B-CN gradually increased up to ~1.47 with increasing $CO_2$ concentration (assuming the interaction of a single $CO_2$

molecule per nitrile group). Such localization of $CO_2$ molecules was driven entirely by the assistance from the high concentration of BA-CN, i.e., ~3 per $Zr_6$-oxo nodes. As seen in Fig. 6d, the overall effect of the solubilization of $CO_2$ increases the local $CO_2$ concentration near the Bi-catalyst to ~0.82 M, ~27-times higher than in bulk aqueous solution (~0.030 M)[24,46]. This value was independently confirmed by electrochemical measurements (Fig. S32, Table S10) using quinone as a redox probe that can reversibly bind with $CO_2$ when reduced[47], as detailed in SI section S2.

Thus, it can be said that the almost insulating UiO-66-B-CN membrane acts as a ≈18.3 µm thick, highly efficient molecular $CO_2$ reservoir, which is grown atop the Bi-foil. Furthermore, the increase and decrease of the $CO_2$ solubilization by successive $CO_2$ and Ar cycles, indicate the dynamic nature of preconcentration of $CO_2$ near Bi-surface by UiO-66-B-CN. Thus, the membrane should be capable of replenishing $CO_2$ from the bulk solution and maintain a high $CO_2$ concentration during electroreduction of $CO_2$. The increased availability of $CO_2$ made a considerable impact on the eCO2RR performance of Bi-UiO-66-CN, increasing both $j_{HCOOH}$ and $FE_{HCOOH}$ compared to Bi-UiO-66-B.

DFT simulations also demonstrate the relation between the basicity of the ligand, computed as the centroid of the N(2p) energies ($E_{N(p)}$ HOMO), for the different interfaces and the respective $CO_2$ adsorption energy ($E_{ads}$ $CO_2$), Fig. 7a. The linear relationship reveals the synergistic effect of UiO-66−CN and Bi. The potential mechanisms for HCOOH production, the competing Hydrogen Evolution Reaction (HER) and alternative paths (towards CO) are presented in Fig. 7b, S31b and S31c, respectively, and were computed using pH=6.8, U = −0.9 $V_{RHE}$ only for the thermodynamic steps. The first route towards HCOOH, M1, involves $CO_2$ adsorption and activation followed by a PCET step to produce HCOOH[6,48]. The alternative, M2, the Heyrovsky mechanism[12], encompasses initial hydride formation on the surface which is transferred to a nearby $CO_2$ (section S1 in SI). The Bi(0001) favors HER and M2 as the energy for H adsorption is −1.12 eV (Fig. S33). As seen in Fig. 7c, for the M1, the $CO_2$ activation is highly endergonic with 2.50 eV (U = −0.9 $V_{RHE}$) for Bi(0001), indicating its unlikelihood to undergo M1, and thus opening up the path towards M2 and HER. This penalty is significantly lowered for Bi-UiO-66 and Bi-UiO-66-CN to 0.20 and −0.10 eV, respectively, hence increasing their likelihood for M1. The energies for subsequent *OCHO intermediate in Bi(0001), Bi-UiO-66,

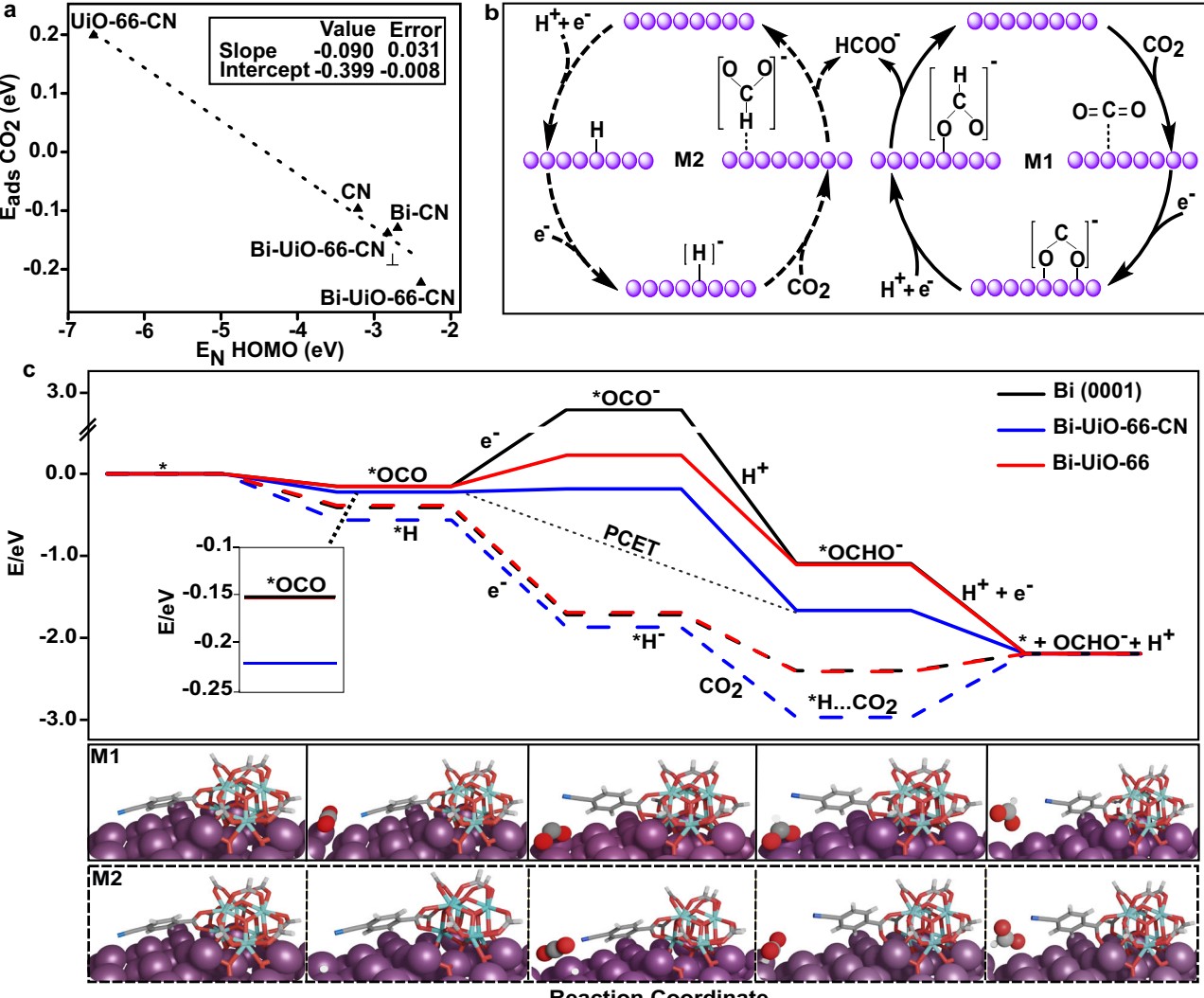

**Fig. 7 | Computational details on eCO2RR pathways predicted for UiO-66-Bi systems. a** DFT computed relation between N(2p) HOMO energy and $CO_2$ adsorption energy ($E_{ads}$ $CO_2$) for different interfaces signifies that the interaction between $CO_2$ and interface improves as the HOMO-Fermi gap of the interface decreases. The synergistic effect of Bi-UiO-66-CN leads to lowest HOMO-Fermi gap and $CO_2$ adsorption energy. **b** Schematic illustration of the 2 eCO2RR pathways towards HCOOH generation as computed with DFT(PBE)-CHE. **c** Reaction energetics for both HCOOH production mechanisms with reaction path illustrations. The solid lines represent HCOOH production via $CO_2$ adsorption and activation followed by a PCET step (M1) and dashed lines represent Heyrovsky mechanism (M2) with initial hydride formation on the surface. Inset highlights the OCO adsorption energy in M1.

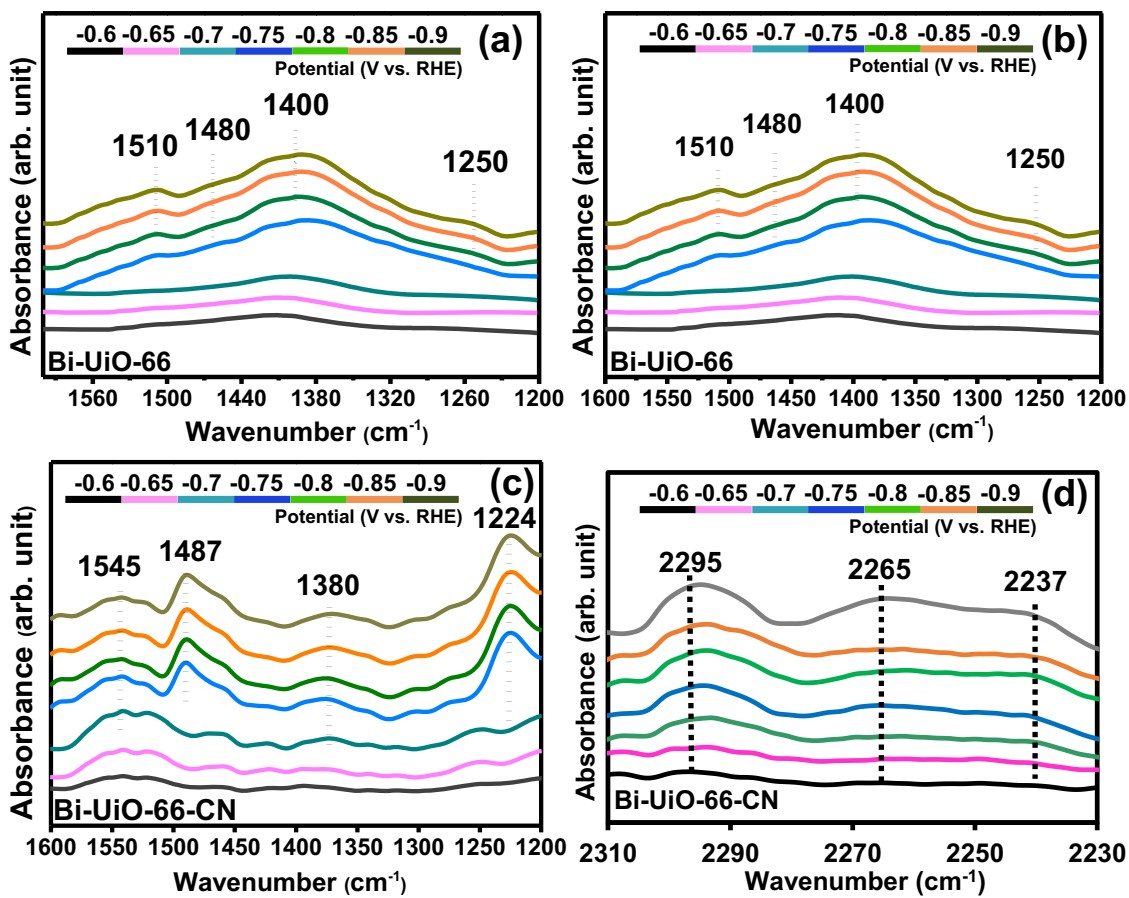

**Fig. 8 | Operando electrochemical ATR-IRRAS analysis.** ATR-IRRAS spectra of **a** Bi-foil, **b** Bi-UiO-66 and **c** Bi-UiO-66-CN under electrochemical $CO_2$ reduction conditions (1200–1600 cm⁻¹). **d** ATR-IRRAS spectrum of Bi-UiO-66-CN in the 2230–2310 cm⁻¹. The ATR-IRRAS spectrum recorded at 0 V was considered as the background and auto-subtracted from all the measurements.

Bi-UiO-66-CN are exergonic with Bi-UiO-66-CN having the lowest energy (−2.90 eV at U = −0.9 $V_{RHE}$) thus best e$CO_2$RR performance. For the M2, the hydride formation and *H···$CO_2$ energies are exergonic for all interfaces making it a downhill process without any energy penalties. The *H···$CO_2$ energy decreases for both Bi-UiO-66 and Bi-UiO-66-CN compared to bare Bi(0001) by −0.02 eV and −0.28 eV, respectively. As for the role of solvation, the reaction profile energetics were computed considering three water molecules explicitly (Fig. S33a). When solvent is considered, *H in Bi-UiO-66 (0.21 eV) and Bi-UiO-66-CN (0.86 eV) are endergonic and thus limit M2 and HER (Fig. S33b), whereas M1 remains feasible.

To gain further insights on factors governing e$CO_2$RR performance in our system, operando ATR-IRRAS electrochemical analysis was carried out for bare Bi, Bi-UiO-66 and Bi-UiO-66-CN. Measurements were recorded at the potential window of −0.6 V to −0.9 V (vs. RHE) with 50 mV intervals (Fig. 8, S34). ATR-IRRAS spectra recorded at 0 V vs. RHE were taken as the background and auto subtracted from all the measurements. For bare Bi and Bi-UiO-66, a set of 4 peaks were observed at applied potentials above −0.7 V vs. RHE (Fig. 7a, b respectively). A broad peak at ~1400–1410 cm⁻¹, corresponding to ν(C − O) band of catalyst-bound *OCHO intermediate (Table S11), and also matches well with earlier reports[49–52]. A peak at 1250 cm⁻¹ assigned to surface adsorbed $HCOOH_{ad}$[53,54], and set of 2 peaks centred around 1500 cm⁻¹, corresponding to carbonate accumulation near the catalytic site[55,56]. Importantly, as seen in Fig. 8c, Bi-UiO-66-CN also exhibits the same 4 distinct IR peaks, albeit to a positional shift of ~30 cm⁻¹ for both *OCHO intermediate (1380 cm⁻¹) and $HCOOH_{ad}$ (1224 cm⁻¹). DFT values showed a similar

behavior, namely, an OCHO* peak shift of 22 cm⁻¹ between bare Bi and Bi-UiO-66-CN (Table S3). To obtain further mechanistic insights, ATR-IRRAS measurements were performed using ¹³$CO_2$, showing an isotopic shift of the ν(C − O) of *OCHO intermediate species (Fig. S35). This ¹³$CO_2$-sensitive vibration (for all the three catalytic systems: Bi, Bi-UiO-66, and Bi-UiO-66-CN) provides direct evidence for the existence of C − O bond of bound *OCHO intermediate species. In turn, it also proves that the relative shift we are seeing in this region under ¹²$CO_2$ atmosphere, i.e., 1400–1410 cm⁻¹ for Bi and Bi-UiO-66, to 1380 cm⁻¹ for Bi-UiO-66-CN, is mainly due to stabilization of the intermediate caused by the Bi-UiO-66-CN membrane (see SI section S2). It is important to mention that $FE_{HCOOH}$ of Bi-UiO-66-CN and the relative area of the 1380 cm⁻¹ peak vary similarly as a function of applied potential (Fig. S36). The striking similarity suggests that the formation and stabilization of *OCHO intermediate dictated the course of e$CO_2$RR and hence significantly impacts product selectivity for Bi-UiO-66-CN. A proposed schematic of the intermediate stabilization by nitrile group onto Bi surface is provided in the inset of Fig. S36a.

Additionally, monitoring the −CN stretching as a function of applied potential (e.g., under catalytic operation), 3 peaks located at 2240 cm⁻¹, 2265 cm⁻¹ and 2295 cm⁻¹, closely match the theoretically obtained ν(C − N) peaks of free −CN (i.e., −CN of UiO-66-CN), $CO_2$-interacted −CN, and *OCHO-interacted −CN respectively. Importantly, the 2295 cm⁻¹ − CN peak has been found to be a suitable probe for the interaction of nitrile with the *OCHO intermediate and further confirms the stabilizing secondary-sphere interactions of the MOF membrane. (Fig. 8d). Similar ν(C − N) frequencies were obtained using DFT at

2238 cm$^{-1}$, 2242 cm$^{-1}$ and 2280 cm$^{-1}$, for free $-CN$, OCO interacted $-CN$, and *OCHO-interacted $-CN$, respectively (Table S4 and section S1 in SI).

As shown by the DFT simulations, M1 is energetically limited by the activation of $CO_2$ on the Bi surface (Fig. 7c). Indeed, extracted Tafel slopes for Bi-UiO-66 and Bi-UiO-66-B-CN were close to the theoretical value of 118 mV/decade (Fig. S36b), which also corresponds to a catalytic rate determining step (RDS) of first electron transfer to activate $CO_2$[7,51,53,57].

Consequently, we conclude that the UiO-66-CN membrane assists eCO$_2$RR in two ways in addition to the factors of attenuated mass transport that was already contributing via the UiO-66 membrane. First, all the BA-CN spread across the UiO-66-CN membrane improves the tri-dimensional $CO_2$ uptake within the MOF-membrane from the aqueous solution, and generates a dynamic $CO_2$ solvation layer near the catalyst, as was observed in the ATR-IRRAS measurement (Fig. 6). Second, at the interface of Bi and UiO-66-CN, it assists $CO_2$ activation as shown in the DFT optimized structure (Fig. 7c) and helps stabilize *OCHO intermediates by secondary sphere interactions.

In this work, a nitrile-functionalized UiO-66 MOF layer (UiO-66-CN) was assembled on a Bi eCO$_2$RR electrocatalyst. In a conventional H-cell measurement setup, the MOF overlayer facilitates substantial enhancement in the catalyst's activity and selectivity, achieving HCOOH faradaic efficiency of up to 93% with 7-times faster kinetics (at −0.75 V vs. RHE). In a GDE configuration current towards HCOOH surpasses the state-of-the-art for this reaction reaching technical application levels (166 mA/cm$^2$). Enhanced catalysis was obtained through i) the MOF acting as a $CO_2$ reservoir in the vicinity of the catalytically active sites, increasing local $CO_2$ concentration by ~27-fold compared to bulk aqueous electrolyte, thus accelerating catalysis; ii) providing secondary-sphere interactions that stabilize HCOOH intermediates (*OCHO) at the catalyst's surface, hence improving electrocatalytic selectivity. Our work presents a means for precise, molecular-level design of efficient heterogeneous eCO$_2$RR electrocatalysts and paves the way for the use of multifunctional materials in these technologies.

## Methods

### The chemicals
Zirconyl dichloride octahydrate (ZrOCl$_2$·8H$_2$O ≥ 99.5%), terephthalic acid (C$_8$H$_6$O$_4$ ≥ 98%), and p-Cyanobenzoic acid (BA-CN, C$_8$H$_5$NO$_2$ ≥ 98%), and Nafion solution perfluorinated resin solution were purchased from Sigma-Aldrich. Acetic acid (AA, CH$_4$CO$_2$, ≥ 98%), dimethylformamide (DMF, C$_3$H$_7$NO), isopropyl alcohol (> 99.8 % AR grade), and NaHCO$_3$ (AR grade) were purchased from Bio-Lab. D$_2$O solution (> 99.9 %) was purchased from Tzamal D-chem laboratories, and Bi-foil (≥ 99%) was purchased from Holland Moran. $CO_2$ and Ar gas (≥ 98%) were purchased from Maxima. SGL 28 BC with Graphene −20% FEPD 121 Filler-Gas diffusion layers (GDL) were purchased from Fuel-CellStore, and $^{13}CO_2$ gas (99% C, < 2% 18 O) was purchased from Cambridge isotope laboratory.

### Pre-treatment of Bi-foil
The Bi-foils were cut into small pieces of size 2 cm$^2$. Each piece of Bi-foil was initially cleaned by polishing using alumina. After polishing, it was cleaned by sonicating in a soap solution, milli-q (mq) water and isopropanol, respectively. These Bi-foils were electrochemically cleaned and activated by performing chronoamperometry at −0.7 V vs RHE for 120 min to remove surface absorbed impurities and convert Bi-oxides present in the surface to Bi. Following that, the Bi electrodes/plates were washed thoroughly with mq. water and dried in Ar flow. The Bi-foils were used to grow the UiO-66 membrane or for electrochemical measurements. Bi-foil's solution-exposed sides could ideally participate in any electrochemical reactions. Thus, the Bi-foil was covered by an insulation tape in such a way that only one face of the Bi-foil was exposed to the electrolyte solution. Of note, the insulation tape showed no electrochemical activity within the potential window of our interest.

### Preparation of the precursor UiO-66 gel
ZrOCl$_2$ (18.6 mg, 0.057 mmol), BDC (10.4 mg, 0.062 mmol), nitric acid (300 µL) and acetic acid (266 µL) was dissolved in 12 mL of DMF by sonication. The clear solution was transferred to a 100 mL glass bottle containing a Teflon cap and kept in a programmable oven at 100 °C for 2 h. The container was cooled to room temperature at the end of the 2 h. Another 12.5 mL of DMF was added to the reaction mixture. Then the already formed gel was made homogeneous by shaking with a vortex shaker. Then the glass bottle was sealed and transferred to a preheated oven at 120 °C for 16 h. Next, another 25 mL of DMF was added to the reaction mixture after cooling it to room temperature. The gel was made homogenous, and the container was kept at 120 °C for another 16 h. We washed the gel to take out excess DMF in the next step. Controlled washing was done with fresh ethanol three times. 50 mL of ethanol was added, and the gel was made homogeneous. The mixture was sealed and transferred to a preheated oven and kept at 60 °C for 24 h. Next, 50 mL of solution was taken out from the top, and another 50 mL of fresh ethanol was added. The mixture was made homogeneous and kept at 60 °C for 24 h. The same step was repeated twice more. The as-synthesized diluted UiO-66-gel was used as the precursor to form the UiO-66 membranes on top of the Bi-foil. Of note, the required volume of the diluted homogeneous precursor UiO-66-gel was different for different thicknesses of the UiO-66 membrane. Each membrane, UiO-66-A-C, was prepared in a single step using the precursor UiO-66-gel. The time required to form the membrane varied in accordance with the volume of the precursor volume (Table S1).

### Preparation of the precursor UiO-66-CN gel by solvent-assisted linker insertion (SALI)
ZrOCl$_2$ (18.6 mg, 0.057 mmol), BDC (10.4 mg, 0.062 mmol), nitric acid (300 µL) and acetic acid (266 µL) was dissolved in 12 mL of DMF by sonication. The clear solution was transferred to a 100 mL glass bottle containing a Teflon cap and kept in a programmable oven at 100 °C for 2 h. The container was cooled to room temperature at the end of the 2 h. The formation of defective UiO-66 gel has already started. Another 12.5 mL of DMF was added to the reaction mixture. Then the gel was made homogeneous by shaking (with a vortex shaker). Then the glass bottle was sealed and transferred to a preheated oven kept at 120 °C for 16 h. Next, 25 mL of DMF was added to the reaction mixture after cooling it to room temperature. The gel was made homogenous by shaking, and the container was kept at 120 °C for another 16 h. In the next step, the gel was modified with BA-CN by solvent-assisted linker insertion (SALI). 100 mg of BA-CN was dissolved in 50 mL of ethanol and was added to the UiO-66-gel, and it was made homogeneous by shaking with a vortex shaker. The mixture was sealed and transferred to a preheated oven and kept at 60 °C for 48 h. Next, 50 mL of solution was taken out from the top, and another 50 mL of fresh ethanol was added. The mixture was made homogeneous and kept at 60 °C for 24 h. The same step was repeated three times. The as-synthesized diluted UiO-66-CN gel was used as the precursor to form the Bi-UiO-66-CN membranes on top of the Bi-foil. Of note, the required volume of the precursor UiO-66-CN gel depends on the desired thickness of the UiO-66-CN membrane. To synthesize the Bi-UiO-66-CN-B membrane, same amount of precursor UiO-66-CN gel was used as that of UiO-66-B, which is 150 µL. Thus, the membrane formation time was also similar, i.e., ≈ 4 h.

### Bi-UiO-66 membrane formation via slow evaporation
The pretreated Bi-foil was used to deposit the Bi-UiO-66 membranes on top of it. Different volumes of UiO-66 gel resulted in the formation of UiO-66 membranes of varying thickness. Please check Table S1 for details. One face of Bi-foil was first coated with insulating tape before coating the thin layer of UiO-66 gel as a precursor. With slow evaporation of the UiO-66 gel over 2–6 h, the UiO-66 membranes were prepared at room temperature (25–30 °C) under a controlled vacuum.

The time required for the formation of the membrane from the gel varies with the thickness of the membrane (Table S1). Synthesis of the Bi-UiO-66-B-CN membrane was done using UiO-66-CN gel, following a similar condition to that of Bi-UiO-66-B.

## Synthesis of Bi-based coordination polymer (Bi-CP) by vapour-assisted conversion (VAC)

Chemglass, Pyrex, 1680, CG-8122, (30 × 60 mm) vials were used for the vapour-assisted conversion (VAC) of Bi-CP onto Bi foil. A mixture of 4.2 mL DMF and 0.8 mL acetic acid was introduced inside the glass bottle/vessel. Then, an electrochemically cleaned Bi-foil was put on top of an inverted 20 mL glass vial kept inside the glass bottle. For the electrochemical cleaning the Bi-foil was treated at −0.9 V (vs. RHE) for 2 h in 0.1 M (aqueous) sodium bicarbonate solution. NaBDC (10.4 mg, 0.062 mmol) and acetic acid (266 μL) were dissolved in 10 mL of DMF. 150 μl of this solution was coated onto Bi-foil. Then the glass bottle/vessel was capped and sealed using Teflon tape before keeping the set-up at 90 °C for 4 h. Because of the acetic acid in the coated solution, Bi foil generates $Bi^{3+}$, which can react immediately with the BDC linker to form the Bi-based coordination polymer. The high surface-area Bi (p-Bi) was synthesized from the Bi-based coordination polymer. Synthesizing the p-Bi electrocatalyst involves two major steps: (1) growth of a Bi-based coordination polymer (Bi-CP) was on top of the Bi-foil; (2) electrochemical conversion of the Bi-CP to p-Bi.

## Electrochemical conversion of Bi-CP to nano-porous Bi (p-Bi)

We have used a conventional three-electrode single compartment cell. Bi-CP, Pt-flag and Ag/AgCl were employed as working, counter, and reference electrodes, respectively. Chronoamperometric measurements were performed at −0.9 V (vs RHE) for 4 h to convert the Bi-CP to p-Bi electrochemically. After the electrolysis, the p-Bi electrode was washed with water and dried in Ar flow. Chronoamperometric conversion was performed at −0.9 V (vs RHE) for 2 h to partially convert the Bi-CP to p-Bi, which was labelled as Bi-CP_half converted. Characterization of Bi-CP, Bi-CP half converted, and p-Bi by XRD and DRIFTS-ATR (Fig. S14) indicated the gradual conversion of coordination polymer to the oxide-containing p-Bi. The p-Bi synthesized by the electrochemical method was later used to grow the porous MOF membrane atop it. So, a set of high surface area Bi-based electrocatalysts were prepared by this method, namely, p-Bi, p-Bi-UiO-66, and p-Bi-UiO-66-CN.

## Synthesis of Bi nanoparticles (NPs) for gas diffusion electrode (GDE)

The Bi NPs were synthesized using the solvothermal route by following the previous report with some modifications[58]. Briefly, the synthesis procedure was as follows: 1.5 mmol of bismuth (III) nitrate pentahydrate and 30 mL ethylene glycol were taken in a 100 mL beaker. Ethylene glycol was used as both a reducing agent and solvent. The solution was sonicated for 5 min to mix the precursors. Then, the dissolved solution mixture was transferred to a 50 mL Teflon vessel, and tightly packed with stainless steel autoclave. The autoclave was kept in a programmable oven for 6 h at 210 °C, and the heating rate of 5 °C/min was used to reach the set temperature. Then, the reaction mixture was cooled to room temperature naturally. Subsequently, the obtained reaction mixture was washed with excess methanol twice. The obtained precipitate was placed in a vacuum oven at room temperature overnight to dry the Bi NPs.

## Fabrication of Bi-GDE and Bi-GDE-UiO-66-CN

First, Bi NPs catalyst ink was prepared. 20 mg of a fine grounded powder of the Bi NPs and 70 μL of Nafion solution were homogeneously dispersed in a mixture of 1:1 ratio of ultra-pure water (1 mL), and isopropyl alcohol (1 mL). Then, the solution was taken to ultra-sonication until it formed homogeneous Bi NPs catalyst ink. To prepare the Bi NPs catalyst on 20% polytetrafluoroethylene (PTFE) coated Gas diffusion layers

(GDL), 300 μL of Bi NPs catalyst ink was drop-cast (1 cm × 1 cm) on GDL electrodes. Subsequently, the electrodes were dried under ambient conditions for ~6 h, and such electrode was termed as Bi NPs coated gas diffusion electrode (Bi-GDE). In all electrochemical experiments, the amount of catalyst loading was maintained at ~3.0 mg/cm². During the electrochemical experiments, the exposed geometrical surface area of the catalyst was maintained 1 cm². To form a UiO-66-CN membrane on dried Bi-GDL, the as-synthesized UiO-66-CN gel (2 mL) was diluted by adding 1 mL of ethanol and homogenized. Thus, diluted homogeneous UiO-66-CN gel was used as a precursor to grow the UiO-66-CN membrane over Bi-GDE. Precisely, 200 μL of diluted UiO-66-CN gel was drop-casted on Bi-GDE. The time required for the formation of the UiO-66-CN membrane on Bi-GDE was ~3 to 4 h at ambient conditions, and such electrode was termed as Bi-GDE-UiO-66-CN. These electrodes were thoroughly characterized using PXRD and SEM measurements (Fig. S24). The XRD pattern of the Bi nanoparticles match well with reference pattern of Bi (JCPDS No. 44-1246) (Fig. S23a). The SEM image indicates a good surface coverage of Bi NPs over the PTFE-C (Fig. S23b). A cross-sectional SEM image was recorded by tilting the GDE and focusing on the cross-section (Fig. S24b).

## Physical characterization methods

The crystalline structures of the UiO-66 thin films were confirmed by PXRD using PAN analytical's Empyream multi-purpose diffractometer instrument and Cu-Kα (0.15405 nm) radiation. The scanning electron microscope (SEM) images were taken using Verios XHR 460 L SEM instrument. Top-view SEM images were recorded after performing carbon coating (5 nm) on the UiO-66 thin film containing Bi-foils. Images were recorded at different magnifications. The SEM-FIB (scanning electron microscopy with focused ion beam) images were recorded using Thermo Scientific Dual-Beam system. The ion beam and the electron intersect at a 52° angle near the sample surface. The samples were first coated with a relatively thicker layer of Au (30 nm) and then analyzed by SEM-FIB. During the analysis, the area under investigation was coated first with 150 nm of Pt and, on top of that, 1500 nm of carbon by ion beam assisted chemical vapor deposition. Next, a small area was cut by the sputtering of the focused ion beam to visualize the cross-sectional thickness of the membrane. ICP-OES analysis was performed using Spectro ARCOS ICP-OES, FHX22 multi-View plasma (SOP, EOP) instrument to determine the amount of Zr in UiO-66 gel. Sample preparation was done by digesting 2 mg of UiO-66 in 5 mL of concentrated $HNO_3$ at 150 °C for 12 h. Later, 1 mL of the solution was diluted to 10 mL for ICP-OES measurement. The X-ray photoelectron (XP) spectra were collected using ESCALAB 250 apparatus X-ray photoelectron spectrometer, with Al-Kα X-ray source and monochromator. The survey spectra were recorded with a pass energy (PE) of 150 eV, and the high energy resolution was achieved with a PE of 20 eV. All measured spectra were calibrated relative to carbon 1 s peak position at 284.6 eV to correct the charging effect, and the data processing was done with the AVANTGE program. The electrochemical measurements were performed using a BioLogic VSP-128 electrochemical workstation. Gas chromatography (GC) was performed by PerkinElmer Clarus 590 GC equipped with a wide-range flame ionization detector (FID), methanizer and thermal conductivity detector (TCD). The head-space gas (after electrolysis) was characterized and quantified by GC measurements. Measurements were performed by manual injection of 100 μL of head-space gas mixture using Pressure Lok Precession Analytical Syringe.

## NMR measurement

Hydrogen Nuclear Magnetic Resonance (H-NMR) measurements were done on the Bruker DPX-500 instrument. A calibration curve of BDC was prepared using standard samples to evaluate the concentration of

BDC and BA-CN in UiO-66 gel and UiO-66-CN, respectively. The different concentration solutions of $H_2BDC$ were prepared in 2 M NaOH/$D_2O$. A known amount of TMACl (tetramethylsilyl chloride) was used as the internal standard for each NMR measurement. A known amount of UiO-66 gel samples was digested in 2 M NaOH/$D_2O$ to determine the amount of BDC and BA-CN.

## DRIFTS-ATR (Diffuse Reflectance Infrared Fourier Transform Spectroscopy)

DRIFTS-ATR were performed on powdered and solid samples. It was done using the Thermo Nicolet iS50 instrument equipped with the Harrick Praying Mantis Diffuse Reflectance Infrared Fourier transform Spectroscopy (DRIFTS) apparatus using MCT-A detector with CdSe window (11700–600 $cm^{-1}$). The MCT-A detector was cooled to liquid $N_2$ temperature before starting the experiments.

## Computational details

Density functional theory (DFT) simulations were performed using Vienna Ab Initio Simulation Package (VASP)[59] version 5.4.4. The functional of choice was PBE[60] with the D3 dispersion by Grimme et al. [61]. Core electrons were represented by Projector Augmented Wave pseudopotentials (PAW)[62] and valence electrons were expanded in plane waves with a kinetic energy cutoff of 450 eV. The Bi-UiO-66-CN model was constructed in three steps by sequential attachment of pre-optimized individual components. First, the bulk Bi (R3m) was optimized, with k-point sampling of $15 \times 15 \times 4$. Then the Bi (0001) surface slab was constructed as a supercell ($5 \times 5$) with 4 layers (lateral size 22.91 Å, thickness 22.91 Å, vacuum ~15 Å) and the k-point sampling was reduced to G-point. The core part of the UiO-66 was independently optimized in a cubic box of 25 Å sides at gamma point. The core of the UiO-66 was capped by H atoms to reduce the computational cost, i.e., the long carboxylates are replaced by formates. For completeness, the UiO-66-CN and the CN linker were also optimized in a similar gas-phase configuration. The gas-phase molecules ($CO_2$, HCOOH, CO, $H_2$, $H_2O$ etc) were relaxed in a cubic box with 20 Å sides. Next, one of the core planes of the UiO-66 with the carboxylate centers are cut to allow O atoms to directly interact with the Bi surface, as this is the most likely type of covalent bond formation. Please see Fig. 3 in the main manuscript. Finally, the −CN linker replaces one of the carboxylate terminations with two different possible orientations to form Bi-UiO-66-CN and was finally optimized. Gamma only k-point sampling and Gaussian smearing with 0.05 eV smearing width was used. Slab calculations included a 15 Å vacuum layer, the dipole correction was applied in z direction and two bottom layers were fixed during the relaxation. Complementarily, three other structures with different possible Bi-UiO-66 linkages were analyzed including OH-Bi, Zr-Bi with excess UiO-66 oxygen atoms bound to Bi at different sites, Zr-Bi (1), and direct Zr-Bi with excess oxygen atoms removed, Zr-Bi (2). The exposed UiO-66 O atoms from the model above were capped with H atoms and allowed to bond with Bi forming an OH-Bi linkage. For the Zr-Bi configurations, the exposed oxygens were either removed from the UiO-66 and directly attached to Bi on different sites, or were completely removed, exposing Zr for a direct Zr-Bi linkage in both cases. All these systems were characterized by Bader charges, XPS, and frequencies for -CN (Table S7). The electronic convergence was at least $10^{-5}$ eV and the force threshold for ionic convergence was set to 0.02 eV/Å. Harmonic frequency calculations were performed using a step size of 0.015 Å and all the atoms except the target molecule were fixed. When considering Stark effects, electric force field of 0.46 eV/Å was applied in z lattice vector (U = 0.90 $V_{RHE}$, pH = 6.8 and d = 3 Å). Electrolyte and solvation effects were included for some frequency calculations and reaction profiles, respectively. The explicit solvation calculations were carried out by placing three water molecules and for the electrolyte cation effect, $Na^+$ (surrounded by two water and a $OH^-$) close the adsorbate. The isotope labelling characterization for $^{12}CO_2$ and $^{13}CO_2$ was done by changing the POMASS from 12 to 13,

respectively. To address the electrochemical problem, the Computational Hydrogen Electron CHE formalism[63] (providing thus only the thermodynamic penalties) was used to obtain the energies of the Proton Coupled Electron Transfer (PCET) steps. The conditions employed for the mechanistic investigation follow the experimental values (pH= 6.8, U = 0.90 $V_{RHE}$ and T = 298 K).

## Electrochemical techniques

Two-compartment cell using a three-electrode set-up was used for the electrochemical measurements. The working compartment consists of the working and reference electrode; on the other hand, the counter electrode was placed in the counter compartment. Both the compartments were separated from each other by a Nafion™ membrane. Unless mentioned otherwise, bare Bi-foil or UiO-66 membrane coated Bi-foil electrodes were used as the working electrode. For high surface area p-Bi, bare p-Bi or UiO-66 membrane coated p-Bi electrodes were used as the working electrode. Ag-AgCl and Pt-flag electrodes were used as reference and counter electrodes, respectively. All electrochemical experiments were performed at ambient temperature, and electrode potentials were converted to the RHE scale using the relation E(RHE) = [E(Ag+/Ag) − (0.059×pH) + 0.197 V]. Linear scan voltammetric (LSV) measurements were recorded at 100 mVs$^{-1}$ scan rate. The data was recorded with a Bio-Logic VSP instrument. $CO_2$ saturated 0.1 M (aqueous) sodium bicarbonate solution was used as the electrolyte for the electrochemical measurements unless mentioned otherwise.

Quinone as redox probe to electrochemically determine $CO_2$ solubility. To probe the $CO_2$ solubilization, the local concentration of $CO_2$ is estimated using the quinone redox couple by employing cyclic voltammetry (CV). CV data were collected for 1 mM 1,4 benzoquinone in an aqueous solution of 0.1 M NaHCO$_3$ under Ar and $CO_2$ atmospheres, respectively. Glassy carbon (GC) and thin layer UiO-66-CN coated GC (GC- UiO-66-CN) were used as working electrodes. Platinum flag and Ag/AgCl (saturated KCl) were used as counter, and reference electrodes, respectively.

## Experimental method for eCO$_2$RR in conventional H-cell

eCO$_2$RR for Bi-flat electrode and high surface-area porous bismuth (p-Bi) electrode with and without the MOF membrane, were performed in gas-tight, $CO_2$ saturated, 0.1 M NaHCO$_3$ containing (pH 6.8), two-compartment cell in a three-electrode configuration. Bi-foil, p-Bi, Bi-UiO-66-A-C, Bi-UiO-66-B-CN, and p-Bi-UiO-66-B-CN were used as the working electrode in different instances. Ag-AgCl and Pt-flag electrodes were used as reference and counter electrodes, respectively. The working compartment and the counter compartment were separated from each other by an ion conducting Nafion™ membrane. The cathodic compartment, i.e., the compartment containing the working electrode, was filled with 35 mL of 0.1 M NaHCO$_3$ solution for electrochemical $CO_2$RR. Separate measurements were done to determine the Faradaic efficiency (FE) for gaseous products ($H_2$ and CO) and liquid products (i.e., HCOOH). To determine FE for $H_2$ evolution (i.e., FE$_{H2}$) and FE for CO (i.e., FE$_{CO}$), $CO_2$ was purged in both the compartments for 1 h, and then the cell was sealed to maintain the $CO_2$ saturation before running chronoamperometry to pass 1 C charge in each case. The gaseous products were collected from the sealed headspace with a Hamilton gas syringe and analyzed by gas chromatographic technique (Agilent). The $CO_2$ RR was carried out for HCOOH quantification while still purging the solution with $CO_2$. Chronoamperometry was performed to pass a total charge of 2 C in each case. At the end of each electrolysis, 10 mL of the electrolyte solution was collected and stored in refrigerator (4−8 °C) of which 400 μL was used for NMR measurements. Faradic efficiencies were calculated based on three sets of NMR measurements for each potential and each sample. The variation in NMR data resulted in a maximum standard deviation of ±3.2% from the reported values. Faradic efficiencies were calculated using the formula: Faradic efficiency (FE) = nF × (m/Q);

where, F= Faradey constant, n = number of electrons, m = number of moles of product formed, Q = total charge passed.

### Experimental method for eCO₂RR in flow-cell mount with GDE

All electrochemical $CO_2$ reduction reactions (eCO₂RR) for Bi-GDE with and without the MOF membranes were carried out using a flow-cell made using Teflon. It consists of two liquid compartments (catholyte and anolyte), and one gas chamber to flow the $CO_2$ gas. Catholyte and anolyte compartments were connected to each other by ion exchange Nafion™ membrane. Chemically inert rubber gaskets were used to separate the chambers and to assemble the cell. For all eCO₂RR, gaseous $CO_2$ was circulated through the gas chamber from the back side of the working electrode (1 cm × 1 cm), and electrolyte of 2.2 M $NaHCO_3$ was circulated through both cathode and anode compartments with the flow rate of 15 mL/min using a pump. Bare GDE, Bi-GDE, Bi-GDE-UiO-66, and Bi-GDE-UiO-66-CN were used as working electrodes in different instances. Ag/AgCl (saturated KCl) and Ni foam were used as reference, and counter electrodes, respectively. During the electrochemical operation, both catholyte and anolyte chambers were supplied with $CO_2$ saturated 2.2 M $NaHCO_3$ electrolyte solution by using a pump with a flow rate of 15 mL/min. CV data were recorded on Bare GDE, Bi-GDE, Bi-GDE-UiO-66, and Bi-GDE-UiO-66-CN in a potential window from 0 V to −1.0 V (vs RHE) at a scan rate of 50 mV/s. For bulk electrolysis, CAs were collected on Bi-GDE, Bi-GDE-UiO-66, and Bi-GDE-UiO-66-CN electrodes at every 0.1 V difference between −0.5 V and −0.9 V (vs RHE), and the same constant charge of 10 C was passed at each potential through the working electrode.

### Stability test of Bi-GDE-UiO-66-CN catalyst

Bulk electrolysis at −0.7 V vs RHE for 25 h was performed for the catalyst Bi-GDE-UiO-66-CN (Fig. S26). $FE_{HCOOH}$ was measured at specific time intervals.

### Electrochemical Impedance Spectroscopy (EIS) characterization

Impedance spectroscopy measurements were carried were carried out in a gas-tight H-cell, in $CO_2$-saturated 0.1 M $NaHCO_3$ solution (pH 6.8), using Ag/AgCl (saturated KCl), Platinum foil, and Bi-foil/Bi-UiO-66/Bi-UiO-66-CN, as the reference, counter, and working electrodes, respectively. For each sample, a set of measurements were done under different applied DC potentials (0.6–0.9 V vs. RHE), by applying a 20 mV AC sinusoidal signal over the constant applied bias with the frequency ranging between 300 kHz and 10 mHz.

### ATR-IRRAS measurement in Otto configuration

ATR measurements were recorded using the Thermo Nicolet iS50 instrument equipped with liquid nitrogen cooled MCT-A detector with CdSe window (11700–600 cm⁻¹). A custom-made spectro-electrochemical cell supplied by the Beijing Scistar technology (Co. Ltd.) was mounded on PIKE instrument VeeMAXIII ATR optical accessory. $CaF_2$ window was used for all the ATR measurements. All the IR data were recorded and processed using the software OMNIC SPECTRA, supplied by Thermo Fischer Scientific. The electrochemical cell consisted of a 3-electrode set-up, where the working electrode sat tightly above the $CaF_2$ window, with the help of an O-ring. The reference and counter electrodes were placed nearby but were not exposed to the incident IR beam. Au-disc and glassy carbon electrodes were used as the working electrodes when and as required. Ag/AgCl electrode was used as the reference electrode, and a Pt foil was employed as the counter electrode for all the measurements. Under specific operational conditions, the working electrode had a Bi or Bi-UiO-66 coating of a few microns thickness. A thin solution layer was present between the Bi-UiO-66 coating and the $CaF_2$ window. The thin layer solution was exposed to the incident IR light. The working electrode surface and the thin layer solution between the IR window and working electrode were studied by the ATR-IRRAS measurement. Analyte or electrolyte solution could be continuously flowed within the cell by solution inlet and outlet using a pump connected to a stock reservoir.

### ATR-IRRAS spectroscopy to determine CO₂ solubility

For $CO_2$ solubility experiments, a gold working electrode was coated with a thin film of UiO-66 membrane or UiO-66-CN membrane as per requirement. 0.1 M $NaHCO_3$ solution continuously flowed through the cell at a constant flow rate of 15 mL/min from a stock reservoir. The stock reservoir was first purged with $CO_2$ and then with Ar for a fixed period. During each purging cycle, a series of ATR measurements were carried out at fixed interval of times. No electrochemical potential was applied. The thickness of the UiO-66 membrane and UiO-66-CN membrane were tuned by coating the UiO-66 gel and UiO-66-CN gel in accordance with the geometrical area of the electrode. ATR spectra were recorded with increasing and decreasing concentrations of $CO_2$. Similarly, $CO_2$ solubilization measurements were performed for an aqueous solution of BA-CN. In this case, the Au disc was not modified with any MOF membrane. 500 mg BA-CN was dissolved in 20 mL of 0.1 M $NaHCO_3$ solution and used as the stock solution for the $CO_2$ solubilization experiment.

### Operando ATR-IRRAS spectroscopy

A similar experimental set-up was used as mentioned above. Instead of an Au-disc working electrode, a glassy carbon electron was used as the working electrode. First, Bi was electrochemically deposited onto a glassy carbon electrode. A conventional three-electrode cell was used along with a 0.1 M $Bi(NO_3)_3$ solution prepared in 0.5 M aqueous $HNO_3$ as electrolyte for the initial deposition of Bi onto a glassy carbon electrode. Pt-flag and Ag/AgCl electrodes were used as counter and reference electrodes, respectively. CV measurements (Fig. S37) were performed to realize the required potential for the Bi deposition. Chronoamperometric measurement was carried out for 2 h at −0.6 V (vs Ag/AgCl) to electrodeposit Bi onto a glassy carbon working electrode. The Bi deposited glassy carbon electrode (Bi-GC) was washed with copious water and dried in Ar flow. The Bi-GC was employed as the working electrode for ATR-IRRAS measurements. For all the ATR-IRRAS measurements, 0.1 M $NaHCO_3$ aqueous electrolyte solution was first purged with Ar followed by purging with $CO_2$ for 20 min before starting the experiments and flowed through the cell with a flow rate of 15 mL/min. Ag/AgCl electrode and Pt-flag electrode were used as the reference and counter electrode. Chronoamperometric measurements were performed for 5 min at each 50 mV interval within the potential range of (−0.6 V vs RHE.) − (−0.9 V vs RHE.) ATR-IRRAS spectra were recorded after 60 s of starting each chronoamperometric measurements. The ATR-IRRAS spectrum recorded at 0 V was considered as the background and auto-subtracted from all the measurements. A similar methodology was used to record the ATR-IRRAS spectra for Bi-UiO-66 and Bi-UiO-66-CN. However, the working Bi-GC was modified with UiO-66 membranes. The same volume of UiO-66 gel, and UiO-66-CN gel were coated onto the Bi-GC to prepare Bi-UiO-66 and Bi-UiO-66-CN, respectively.

## Data availability

The data that support the findings of this study are available from the corresponding author upon request. Source data of the spectroscopic and other characterizations, electrochemical measurements and theoretical calculations for the main text are provided with this paper. Source data are provided with this paper.

## Code availability

The calculation files for all the structures can be inspected through ioChem-BD database at this link: https://doi.org/10.19061/iochem-bd-1-303.

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

## Acknowledgements

We thank the Ilse Katz Institute for Nanoscale Science and Technology for the technical support of material characterization. The project leading to this application has received funding from European Research Council (ERC) under the European Union's Horizon 2020 research and innovation programme under grant agreement No. 947655. This work was partially funded by the Israel Science Foundation (ISF) (grant No. 1267/22). S. M. is grateful for a Kreitman postdoctoral fellowship, R.S. and I. L. thank Kreitman Ph.D. fellowship. MS.N. and N.L. thank the Barcelona Supercomputing Center (BSC-RES) for providing computational resources and Ministerio de Ciencia e Innovación, with Ref. No. PID2021-122516OB-I00 for the supporting this work.

## Author contributions

S.M. performed most of the experiments, analyzed the data, and wrote the initial draft of the manuscript. MS.N. performed the theoretical calculations. G.S.S. performed the electrochemical measurements including the GDE experiments, and analyzed the data. A.R.K., and M.S., helped with the GDE set-up and experiments. A.G. performed various electrochemical measurement, which includes the operando ATR-IRRAS measurements. MS.N., A.G., and G.S.S., helped with the manuscript writing. R.S., I.L., R.I., helped with ATR-IRRAS, DRIFT-IR measurement, and PXRD, ICP, and SEM analysis. L.A. developed the NMR program used for HCOOH quantification. R.S., I.R., helped with the NMR measurements. R.B. helped with the synthesis. N.L., and I.H. supervised the project and co-wrote the paper. All the authors discussed the results and reviewed the manuscript.

## Competing interests

The authors declare no competing interests.
