## [Peer Review File · Nature Communications]

REVIEWER COMMENTS

Reviewer #1 (Remarks to the Author):

In the revised paper, the reviewer's comments are adequately answered. I believe that the manuscript is ready for publication.

Reviewer #3 (Remarks to the Author):

This work reports a film of nitrile-modified metal-organic framework (MOF) covered on catalysts to act as CO₂ reservoir and promote electrochemical CO₂ reduction. Although many supplementary experiments have been added according to the reviewers' comments from Nat. Catal, there are still many key evidences and inferences that still have some irrationalities. On account of this, major revision is required at least to further improve the quality of this work and it should be reevaluated after addressing the following concerns:

1. The linear sweep voltammetry (LSV) curves of Bi, Bi-UiO-66 and Bi-UiO-66-CN in Ar-saturated electrolyte should also be compared to show the influence of MOF layers.
2. The ¹³CO₂ should also be employed to confirm the increase of CO₂ solubility by ATR-IRRAS spectrum according ref. 24.
3. The differences in CO₂ adsorption behavior and enthalpy between UiO-66-CN and UiO-66 should also be presented by CO₂ adsorption isotherms.
4. It is still hard to identify whether -CN promotes CO₂ solvation or directly bonds to CO₂, which seems to be two different concepts. The conception of solvated CO₂ and the CO₂ binding with -CN should be clarified. Meanwhile, the authors are suggested to give detailed explanations on the promotion effect of -CN to boost the solvation of CO₂ in NaHCO₃.
5. From the infrared analysis results, the degree of CO₂ solvation and the relative peak intensity of CO₂(aq) after the addition of BA-CN molecules are even higher than that of UiO-66-CN. Moreover, from the calculation results, only -CN groups in close proximity to Bi play the significant role. What is the superiority of such a thick MOF membrane compared with BA-CN molecules on electrochemical CO₂ reduction?
6. The local CO₂ concentration near the Bi-catalyst after the introduction of BA-CN molecules should also

be calculated and compared with that of UiO-66-CN

7. The feasibility of the method using quinone as redox probe to determine CO₂ solubility is doubtful. As is shown in ref. 47, organic electrolyte or non-protonic solvent is employed in such method rather than aqueous solution. As a matter of fact, the quinone is much easier to bind protons rather than to bind CO₂ in aqueous solution.

8. The authors claim that the insulate MOF membrane could enhance current density without mass transfer limitation. To prove this, EIS spectra of Bi, Bi-UiO-66 and Bi-UiO-66-CN under different applied potentials are suggested to be given and analyzed.

9. In Figure 7c, there are almost no energy barriers for the reduction of CO₂ to *OCHO⁻ via 2e⁻ and 2H⁺ transfer, which is difficult to understand.

10. The authors should explain why UiO-66 without -CN can also significantly improve the Faraday efficiency of HCOOH, which is more evident than -CN. How to exclude such enhancement mechanism in UiO-66 from UiO-66-CN?

Response to Reviewers' Comments:

Reviewer #1:

In the revised paper, the reviewer's comments are adequately answered. I believe that the manuscript is ready for publication.

We thank the Reviewer for the positive assessment of our work and accepting it for publication in Nature Communications.

Reviewer #3:

This work reports a film of nitrile-modified metal-organic framework (MOF) covered on catalysts to act as CO₂ reservoir and promote electrochemical CO₂ reduction. Although many supplementary experiments have been added according to the reviewers' comments from Nat. Catal, there are still many key evidences and inferences that still have some irrationalities. On account of this, major revision is required at least to further improve the quality of this work and it should be reevaluated after addressing the following concerns:

We thank the Reviewer for the comments. We have responded to the comments of the Reviewer in a point-by-point manner. In the following, the Reviewer's comments are in black, our answers in blue and the actions taken in bold. We have incorporated modifications in the manuscript and supporting and marked it with yellow background.

1. The linear sweep voltammetry (LSV) curves of Bi, Bi-UiO-66 and Bi-UiO-66-CN in Ar-saturated electrolyte should also be compared to show the influence of MOF layers.

We have performed the LSV experiments for Bi, Bi-UiO-66 and Bi-UiO-66-CN and added it in the modified supplementary information (Figure S10) and mentioned it in the main text (paragraph 3, Page 7). The MOF membrane coated Bi electrodes show lower catalytic current for hydrogen evolution compared to bare Bi foil, which indicates that the MOF membranes suppress the hydrogen evolution reaction on the catalyst surface (Figure 1 in response letter).

Figure 1. Linear sweep voltammetry of Bi, Bi-UiO-66 and Bi-UiO-66-CN under Ar in 0.1 M NaHCO₃. Scan rate was 100 mV/s.

2. The $^{13}\text{CO}_2$ should also be employed to confirm the increase of CO_2 solubility by ATR-IRRAS spectrum according ref. 24.

In order to confirm the CO_2 solubility peak, operando electrochemical study was performed using isotope labelled $^{13}\text{CO}_2$. Both UiO-66-B and UiO-66-B-CN membrane coated Bi electrodes were subjected to ATR-IRAAS analysis in 0.1 M NaHCO_3 aqueous solution after purging with $^{13}\text{CO}_2$ gas for 20 minutes at 0V vs. RHE, i.e. the potential where CO_2 reduction does not take place. This methodology ensures that the solubility of CO_2 can be determined without perturbation. It has been found with UiO-66-B (Figure 2 in response letter, red spectra) a broad peak at 2290 cm^{-1} is generated, which corresponds to the gaseous $^{13}\text{CO}_2$ peak along with a very small peak at 2270 cm^{-1} corresponding to dissolved $^{13}\text{CO}_2$ in water. For UiO-66-B-CN, the peak at 2290 cm^{-1} is sharp. Instead, the peak at 2270 cm^{-1} is higher in intensity compared to that for UiO-66, indicating higher solvation of $^{13}\text{CO}_2$ in presence of $-\text{CN}$ group. The corresponding 70 cm^{-1} $^{12/13}\text{CO}_2$ shift is in well accordance with the previous literature report (*Anal. Methods*, 2016, 8, 756-762) and therefore **confirms the increase in CO_2 solubility in presence of $-\text{CN}$ without doubt. We have added the figure in the modified Supplementary Information (Figure S28) along with a brief discussion in main text (paragraph 2, Page 10).**

Figure 2. ATR-IRAAS spectra of UiO-66-B(red) and UiO-66-B-CN (blue) coated Bi electrode at 0V vs. RHE under operando condition in presence of $^{13}\text{CO}_2$ -saturated 0.1 M NaHCO_3 aqueous solution in the CO_2 region.

3. The differences in CO_2 adsorption behavior and enthalpy between UiO-66-CN and UiO-66 should also be presented by CO_2 adsorption isotherms.

In this manuscript, we are already showing by two independent techniques that the UiO-66-CN membrane forms a dynamic CO_2 solvation layer that increases the local CO_2 concentration. To be more specific, we have reported increased CO_2 solubilization by the MOF membrane dipped inside aqueous bicarbonate solution and we do not deal with the gas phase CO_2 sorption by the MOF, either for the H-cell or for the GDE equipped flow-cell. We have proved the enhancement of local CO_2 concentration compared to the bulk aqueous solution by using IR spectroscopy, and electrochemical measurements using quinone based redox

probe. Therefore, we consider that studying the gas phase CO₂ sorption would only marginally contribute to the discussion, as gas phase measurements correspond to different conditions from the ones of the presented work, which deals with solubility enhancement in aqueous bicarbonate solution.

4. It is still hard to identify whether -CN promotes CO₂ solvation or directly bonds to CO₂, which seems to be two different concepts. The conception of solvated CO₂ and the CO₂ binding with -CN should be clarified. Meanwhile, the authors are suggested to give detailed explanations on the promotion effect of -CN to boost the solvation of CO₂ in NaHCO₃.

Indeed, there are two different ways in which UiO-66-CN membrane helps catalysis. First, it improves the tri-dimensional CO₂ uptake within the MOF-membrane, thus increasing the local CO₂ concentration, as was demonstrated by the ATR-IRRAS measurements (Figure 6). Second, at the Bi-UiO-66-CN interface, it improves CO₂ activation as shown in the operando ATR-IRRAS analysis (Figure 8d), and the DFT optimized structure (Figure 7c).

Specifically, the CO₂ solubility enhancement of UiO-66-CN MOF membrane was confirmed by: (a) the reversibly growing peak at $\approx 2342\text{ cm}^{-1}$ in ATR-IRRAS (in Otto configuration) spectra of UiO-66-CN membrane recorded during CO₂-purge cycle (Figure 6); (b) comparing the ΔE (difference in $E_{1/2}$ values in Ar and CO₂ atmosphere) for glassy carbon electrode and UiO-66-CN membrane coated glassy carbon electrode using quinone based redox probe, which binds reversibly with CO₂ (Figure S30 and Table S9).

As for confirmation of CO₂ activation at the catalyst-MOF interface, detection of the peak at 2265 cm^{-1} in operando ATR-IRRAS analysis of UiO-66-CN in presence of CO₂, confirm the interactions between the -CN functionality and dissolved CO₂. The ATR-IRRAS analysis was further supported by the DFT calculations.

We have clarified these two aspects in the main text (paragraph 3, Page 15).

5. From the infrared analysis results, the degree of CO₂ solvation and the relative peak intensity of CO₂(aq) after the addition of BA-CN molecules are even higher than that of UiO-66-CN. Moreover, from the calculation results, only -CN groups in close proximity to Bi play the significant role. What is the superiority of such a thick MOF membrane compared with BA-CN molecules on electrochemical CO₂ reduction?

We would like to clarify that the simulations only address the mechanism of CO₂ activation at the Bi-UiO-66-CN interface and not with the tri-dimensional solubility increase.

Using aqueous solution of BA-CN would remove the advantages of heterogeneous CO₂ solvation layer and introduce limitations because of dissolved BA-CN in the solution.

Instead, as answered to comment (4), the role of the UiO-66-CN membrane is two-folded, meaning: i) to increase local CO₂ concentration, and ii) to improve activation (as also identified by the DFT simulations). Specifically, the UiO-66-CN membrane with BA-CN improves the eCO₂RR performance by two mechanisms: (1) all the BA-CN moieties spread across the UiO-66-CN membrane help solubilize CO₂; (2) the BA-CN molecules near the Bi surface stabilizes the eCO₂RR intermediate (*OCHO) (identified by the DFT simulations). Thus, the whole UiO-

66-CN membrane behaves as a $\approx 18 \mu\text{M}$ thick CO_2 solvation layer which also helps stabilize eCO_2RR intermediate. **Both the factors are crucial for the eCO_2RR .**

This has been added to the manuscript (paragraph 3, Page 15).

6. The local CO_2 concentration near the Bi-catalyst after the introduction of BA-CN molecules should also be calculated and compared with that of UiO-66-CN

As mentioned in the response to comment (4) and (5), the CO_2 purging experiment in the aqueous solution containing BA-CN was done as a control study to qualitatively relate the variation in IR spectra with that of UiO-66-CN under similar condition. As detailed in the manuscript, this experiment helped to better understand the effect of immobilized BA-CN on CO_2 solubility in case of UiO-66-CN. The eCO_2RR was performed using Bi as a catalyst and the UiO-66-CN membrane functioned as a CO_2 solvation layer. We have not quantified the CO_2 solubility for the BA-CN containing homogeneous aqueous solution as it would not provide any information that can be necessary for the eCO_2RR and more relevant than the information obtained. More importantly, the effect of homogeneous aqueous solution of BA-CN to increase CO_2 solubility may not be quantitatively similar to the effect of molecularly tethered BA-CN in the UiO-66-CN membrane. Such quantitative comparison can be an independent study by itself and is beyond the scope of the presented work. Thus, the authors do not think that quantifying the CO_2 solubility enhancement by homogeneous BA-CN solution is necessary and helpful for the presented work in the manuscript.

7. The feasibility of the method using quinone as redox probe to determine CO_2 solubility is doubtful. As is shown in ref. 47, organic electrolyte or non-protonic solvent is employed in such method rather than aqueous solution. As a matter of fact, the quinone is much easier to bind protons rather than to bind CO_2 in aqueous solution.

We would like to clarify the feasibility of the experimental method for determining CO_2 concentration using quinone as the probe. First, ref. 47 does not show any quinone related experiments. In that paper, the electrocatalytic CO_2 reduction by a Mn-bpyridine catalyst is shown in organic solvent in presence of a source of proton. Quinones are well-studied examples of organic redox couples in aqueous and non-aqueous electrolyte solutions (*J. Am. Chem. Soc.* **2007**, *129*, 12847-12856). It is well documented in the aqueous solution, quinones are electrochemically easily reduced to hydroquinones by consuming protons from the electrolyte solution (2H^+ , 2e^-) under Ar atmosphere, which produces well-defined symmetrical redox couple in cyclic voltammetry (CV). However, the redox properties of quinones were also investigated in aqueous solution in presence of saturated CO_2 atmosphere. Under such conditions, in the electrochemical reduction process of quinones, CO_2 gets bound to its anionic form along with protons which increases ΔE_p (peak to peak separation) along with peak width at half maxima and also shifts cathodic and anodic peak potentials individually in the CV data of quinone compared to that in Ar atmosphere (*Nat. Commun.* **2020**, *11*, 2278). From the obtained ΔE_p values in the CV data of quinone in saturated CO_2 vs. Argon, the concentration of bound/adsorbed CO_2 on the catalyst surface can be determined according to the equation given in ref. 47. This concept has also been further used for carbon capture and release experiments using an electrochemical approach (*MRS Commun.* **2023**, *13*, 994-1008). Similarly, in our case for both electrodes (with and without UiO-66-CN), under CO_2

atmosphere, ΔE_p increases in CO_2 atmosphere compared to that in Ar atmosphere for quinone redox couple, albeit to the fact that the increase in ΔE_p is larger by 0.043V for UiO-66-CN MOF (see Figure 3 in response letter). This clearly suggests, that the exposed -CN functional group of UiO-66-CN has a role in the enhancement of CO_2 binding/adsorption at the electrode. Detailed calculations are included in the supporting information.

We have further added a discussion on the applicability of the electrochemical method using quinone as redox probe to quantify relative CO_2 concentration in the Supplementary Information.

Figure 3. (a) and (b) cyclic voltammogram (CVs) of 1 mM 1,4 benzoquinone on GC, and UiO-66-CN-GC in 0.1M NaHCO_3 electrolyte solution under Ar and CO_2 atmosphere, respectively.

8. The authors claim that the insulate MOF membrane could enhance current density without mass transfer limitation. To prove this, EIS spectra of Bi, Bi-UiO-66 and Bi-UiO-66-CN under different applied potentials are suggested to be given and analyzed.

As suggested, we have performed an EIS measurement of Bi, Bi-UiO-66 and Bi-UiO-66-CN under different applied potentials. In all samples, we have detected the existence of 2 RC time-constants. A faster RC attributed to charge transfer at the Bi-electrolyte interface, as well as a slower RC, corresponding to diffusional mass-transport in solution (*Energy Rep.* **2022**, *8*, 7964–7975). We have analyzed the data and plotted both R_{ct} (charge transfer resistance) and R_{mt} (mass transport resistance) as a function of applied potential (see figure 4 in response letter). Regarding R_{ct} , one can observe that charge transfer resistance for Bi foil is larger than for Bi-UiO-66 and Bi-UiO-66-CN, thus signalling for acceleration of catalysis rate (e.g. higher catalytic currents) for the MOF-coated samples (Figure 4a). As for R_{mt} , we see that Bi-UiO-66 exhibits the highest mass transport resistance, while both Bi foil and Bi-UiO-66-CN have similar R_{mt} values. Meaning, compared to bare Bi foil, UiO-66 coated Bi shows higher resistance for diffusional mass transport in solution (in accordance with attenuation of diffusional mass transport through the UiO-66 membranes). Yet, UiO-66-CN coated Bi exhibits similar mass transport resistance as Bi foil, presumably due to the high local CO_2 concentration within the UiO-66-CN membrane, thus enabling larger reactant delivery toward the catalytic surface (Figure 4b). the existence of MOF-membrane coating on the Bi foil surface does not significantly limit diffusional mass transport of ions and catalytic-substrates toward the catalytically-active sites.

We have added the EIS results to the Supporting Information (Figure S11) and discussed these experiments in the main text (paragraph 4, Page 7) and Supplementary Information.

Figure 4. Electrochemical Impedance spectroscopy (EIS) analysis of Bi, Bi-UiO-66 and Bi-UiO-66-CN under different applied potentials, (a) R_{ct} (charge transfer resistance), (b) R_{mt} (mass transport resistance), and (c) equivalence circuit used for fitting impedance data.

9. In Figure 7c, there are almost no energy barriers for the reduction of CO_2 to $^*\text{OCHO}$ - via $2e^-$ and 2H^+ transfer, which is difficult to understand.

The Computational Hydrogen Electrode (*J. Phys. Chem. B* **2004**, 108, 17886–17892) cannot provide barriers as it is a thermodynamic model, but it has been at the core of the understanding of electrochemical processes since its inception. As we are using a $U = -0.9 \text{ V}_{\text{RHE}}$ all the thermodynamic steps shall be downhill which is exactly what the model certifies. **To clarify this aspect, we have added a comment to the main text (paragraph 1, Page 12), in the caption of the Figure 7, and in Supporting Information (Page S12).**

10. The authors should explain why UiO-66 without -CN can also significantly improve the Faraday efficiency of HCOOH , which is more evident than -CN. How to exclude such enhancement mechanism in UiO-66 from UiO-66-CN?

As already mentioned in the manuscript, first the UiO-66 membrane thickness was varied and optimized to achieve best $e\text{CO}_2\text{RR}$ performance. The variation of thickness caused variation of $e\text{CO}_2\text{RR}$ performance due to mass transport variation, which was demonstrated in our earlier work (*Angew. Chem. Int. Ed.* **2021**, 60, 13423–13429). Hence, the UiO-66-B possess the optimized effect of mass transport through the MOF membrane. As mentioned in the manuscript, after optimization the membrane thickness, UiO-66-CN membrane was developed with similar thickness to that of UiO-66-B. This strategy was adapted to incorporate

the effect of CO₂ solvation and reaction intermediate stabilization by the BA-CN on top of the factor that was already contributing. The observed eCO₂RR performance of the Bi-UiO-66-CN was a result of all the contributing factors.

A clarification of this aspect was added in the manuscript (paragraph 3, Page 15).

REVIEWER COMMENTS

Reviewer #3 (Remarks to the Author):

Most of the comments in the former round of review have been addressed by the authors. Now, I have only one more question and once it is clarified, I would recommend the publication of this work. The specific comment is listed below:

The peaks at 2345 cm⁻¹ in Figure 6 show the enhanced CO₂ solubility in aqueous solution after coating UiO-66-CN on Bi. Since the chemical binding between –CN and CO₂ is occurred at 2270 cm⁻¹, it is hard to understand –CN groups can further enhance CO₂ solubility in solution. The chemical mechanism behind such enhancement effect should be discussed in detail. Moreover, it is puzzling that the ¹³CO₂ labelling experiments are performed at 0 V vs RHE. It should be performed under same conditions as the conditions in Figure 6a for better comparison.

Response to Reviewers' Comments:

Reviewer #3:

Most of the comments in the former round of review have been addressed by the authors. Now, I have only one more question and once it is clarified, I would recommend the publication of this work. The specific comment is listed below:

We thank the Reviewer for the positive assessment of our manuscript.

The peaks at 2345 cm⁻¹ in Figure 6 show the enhanced CO₂ solubility in aqueous solution after coating UiO-66-CN on Bi. Since the chemical binding between -CN and CO₂ is occurred at 2270 cm⁻¹, it is hard to understand -CN groups can further enhance CO₂ solubility in solution. The chemical mechanism behind such enhancement effect should be discussed in detail. Moreover, it is puzzling that the ¹³CO₂ labelling experiments are performed at 0 V vs RHE. It should be performed under same conditions as the conditions in Figure 6a for better comparison.

The CN groups enhance CO₂ solubility not in solution **but within the MOF membrane with respect to solution.**

The chemical mechanism implies that CO₂ interacts with the CN groups via the donor nature of the HOMO in the CN group. This is **shown in the dependence of the CO₂ interaction with different donors as shown in Figure 7a (also attached below), where the highest the HOMO for the ligand the higher the interaction with CO₂.** We want to clarify again that this is not exclusive to the layer close to the Bi surface but also to the tridimensional structure of the material.

As for the suggested experiment, severe problems with ¹³CO₂ supplies do not allow us to carry out the extra experiments suggested. However, in any case, these experiments will not change the conclusions of all the detailed investigations that we have already presented.

REVIEWERS' COMMENTS

Reviewer #3 (Remarks to the Author):

The authors have almost addressed our concern. I therefore recommend the publication of this paper in Nat. Commun.